# SARS-CoV-2 spike protein induces inflammation via TLR2-dependent activation of the NF-κB pathway

**Shahanshah Khan[1], Mahnoush S Shafiei[1], Christopher Longoria[2], John W Schoggins[3], Rashmin C Savani[2], Hasan Zaki[1]\***

[1]Department of Pathology, The University of Texas Southwestern Medical Center, Dallas, United States; [2]Department of Pediatrics, The University of Texas Southwestern Medical Center, Dallas, United States; [3]Department of Microbiology, The University of Texas Southwestern Medical Center, Dallas, United States

**Abstract** The pathogenesis of COVID-19 is associated with a hyperinflammatory response; however, the precise mechanism of SARS-CoV-2-induced inflammation is poorly understood. Here, we investigated direct inflammatory functions of major structural proteins of SARS-CoV-2. We observed that spike (S) protein potently induced inflammatory cytokines and chemokines, including IL-6, IL-1β, TNFα, CXCL1, CXCL2, and CCL2, but not IFNs in human and mouse macrophages. No such inflammatory response was observed in response to membrane (M), envelope (E), and nucleocapsid (N) proteins. When stimulated with extracellular S protein, human and mouse lung epithelial cells also produced inflammatory cytokines and chemokines. Interestingly, epithelial cells expressing S protein intracellularly were non-inflammatory, but elicited an inflammatory response in macrophages when co-cultured. Biochemical studies revealed that S protein triggers inflammation via activation of the NF-κB pathway in a MyD88-dependent manner. Further, such an activation of the NF-κB pathway was abrogated in Tlr2-deficient macrophages. Consistently, administration of S protein-induced IL-6, TNF-α, and IL-1β in wild-type, but not Tlr2-deficient mice. Notably, upon recognition of S protein, TLR2 dimerizes with TLR1 or TLR6 to activate the NF-κB pathway. Taken together, these data reveal a mechanism for the cytokine storm during SARS-CoV-2 infection and suggest that TLR2 could be a potential therapeutic target for COVID-19.

**\*For correspondence:** hasan.zaki@utsouthwestern.edu

## Editor's evaluation

Your paper provides an important advance regarding the role of SARS-CoV-2 spike protein as an inducer of the innate inflammatory cascade in both epithelial cells and macrophages. This is an important early event in development of the cytokine storm associated with COVID-19 and may be of therapeutic relevance.

## Introduction

Coronavirus induced disease (COVID)19, caused by severe acute respiratory syndrome coronavirus 2 (SARS-CoV-2), has emerged as a major public health crisis since December 2019 (*Zhou et al., 2020b*; *Wu et al., 2020*; *Huang et al., 2020*). SARS-CoV-2 is a positive sense single-stranded RNA virus. Like other coronaviruses, such as SARS (retrospectively named SARS-CoV-1) and Middle Eastern respiratory syndrome (MERS)-CoV, SARS-CoV-2 primarily causes infection in the respiratory tract, leading to either asymptomatic infection or a range of symptoms including cough, fever, pneumonia, respiratory failure, along with other complications like diarrhea and multi-organ failure (*Vabret et al., 2020*; *Tay*

*et al., 2020*; *Li et al., 2020*; *Gu et al., 2005*). In the absence of effective therapies, COVID-19 has caused over 4 million deaths worldwide by the end of 2021. Although our knowledge is still evolving, immunopathology caused by the cytokine storm plays a decisive role in COVID-19 pathogenesis (*Vabret et al., 2020*; *Tay et al., 2020*; *Blanco-Melo et al., 2020*).

SARS-CoV-2 infects human cells through its Spike (S) protein, which binds to the receptor angiotensin-converting enzyme 2 (ACE2), expressed on alveolar epithelial cells, allowing endocytosis of the viral particle (*Zhou et al., 2020b*; *Hoffmann et al., 2020*). Following endocytosis, the viral genome is replicated using both viral and host machineries, leading to the death of virally infected cells (*Walls et al., 2020*). The pathology of SARS-CoV-2 infected lung is further worsened with inflammatory responses of innate immune cells such as macrophages, monocytes, and neutrophils, which are activated by viral components and products of apoptotic and necrotic cells (*Huang et al., 2020*). While the innate immune response is essential for antiviral host defense, excessive inflammatory cytokines and chemokines are cytotoxic for respiratory epithelial cells and vascular endothelial cells (*Xu et al., 2020*). Indeed, the clinical manifestation of COVID-19 is marked by higher concentrations of IL-2, IL-6, IL-8, TNFα, IFNγ, MCP1, MIP1α, IP-10, and GMCSF in patients' blood (*Huang et al., 2020*; *Wang et al., 2020*; *Mehta et al., 2020*). Disease severity and death of COVID-19 patients have been correlated to the elevated expression of IL-6 and TNFα (*Huang et al., 2020*; *Hadjadj et al., 2020*). However, our understanding of the precise mechanism of the induction of proinflammatory cytokines and chemokines during SARS-CoV-2 infection is very limited.

The innate immune inflammatory response is initiated with the recognition of pathogen-associated molecular patterns (PAMPs) by pattern recognition receptors (PRRs), such as Toll-like receptors (TLRs), NOD-like receptors (NLRs), and RIG-I like receptors (RLRs). Activated PRRs involve multiple signaling adapters to activate transcription factors, such as NF-κB, AP1, and IRF3, which regulate the expression of genes involved in immunity and inflammation. RNA sensing receptors such as TLR7, RIG-I, and MDA5 play a central role in antiviral immunity by inducing type I interferons (IFNα and IFNβ) via IRF3 and NF-κB (*Khan et al., 2019*; *Park and Iwasaki, 2020*; *Kawasaki and Kawai, 2014*; *Kawai and Akira, 2006*; *Jensen and Thomsen, 2012*). Although the relative contribution of RNA sensing pathways in SARS-CoV-2-mediated immunopathology is yet to be explored, previous studies reported that macrophages and dendritic cells infected with SARS-CoV-1 produce proinflammatory cytokines and chemokines, but not type I interferons (*Cheung et al., 2005*; *Law et al., 2005*). Consistently, inflammatory responses in severe COVID-19 patients are characterized by high levels of proinflammatory cytokines, but poor type I interferon response (*Blanco-Melo et al., 2020*; *Hadjadj et al., 2020*). Beyond these phenotypic observations, the precise mechanism of the hyperinflammatory response during SARS-CoV-2 infection is poorly understood.

SARS-CoV-2 is an enveloped virus consisting of four major structural proteins—S, nucleocapsid (N), membrane (M), and envelop (E) (*de Wit et al., 2016*). S protein binds to the receptor-binding domain of ACE2 through its S1 subunit allowing proteasomal cleavage of S protein and fusion of the S2 subunit with the host cell membrane (*Hoffmann et al., 2020*; *Walls et al., 2020*; *Li et al., 2003*; *Bertram et al., 2011*; *Shirato et al., 2011*). Thus, SARS-CoV-2 structural proteins are likely to be exposed to PRRs located on the cell membrane, endosome, and cytosol of the infected cell. However, our knowledge of the role of SARS-CoV-2 structural proteins in the innate immune response is very limited. Here, we investigated the inflammatory properties of S, M, N, and E proteins and revealed that S protein, but not M, N, and E proteins, is a potent viral PAMP, which stimulates macrophages, monocytes, and lung epithelial cells. We demonstrated that S protein is sensed by TLR2, leading to the activation of the NF-κB pathway and induction of inflammatory cytokines and chemokines. This study provides critical insight into the molecular mechanism that may contribute to cytokine storm during SARS-CoV-2 infection.

## Results

### SARS-CoV-2 S protein induces inflammatory cytokines and chemokines in macrophages and monocytes

Macrophages play a central role in the hyperinflammatory response during SARS-CoV-2 infection (*Grant et al., 2021*). To understand whether SARS-CoV-2 structural proteins can activate macrophages and monocytes, we stimulated human monocytic cell THP1-derived macrophages with

recombinant S1, S2, M, N, and E proteins. Interestingly, both S1 and S2 proteins induced proinflammatory cytokines *IL6*, *TNFA*, and *IL1B*, with S2 being more potent, as measured by real-time RT-PCR and ELISA (*Figure 1A* and *Figure 1—figure supplement 1A*). Chemokines produced by macrophages and monocytes recruit T lymphocytes and other immune cells in the inflamed tissue, aggravating inflammatory damage (*Vabret et al., 2020*; *Grant et al., 2021*). Both S1 and S2 subunits of S protein-induced chemokines including *CXCL1, CXCL2,* and *CCL2* in THP1 cells (*Figure 1A*). Interestingly, three other structural proteins—M, N and E—did not induce any cytokines and chemokines (*Figure 1A*). S1 and S2 exerted a synergistic effect on cytokine production when cells were incubated with them together (*Figure 1—figure supplement 1B*). However, other structural proteins, including M, N, and E, had no effect on S-induced inflammatory responses (*Figure 1—figure supplement 1B*). Interferons are critical for adaptive immune response and antiviral immunity (*Park and Iwasaki, 2020*). However, THP1 cells did not express either type-I (*IFNA* and *IFNB*) and type-II (*IFNG*) interferons in response to any of the SARS-CoV-2 structural proteins (*Figure 1A*). Notably, THP1 cells are not defective in producing interferons when activated by PolyI:C, a ligand for TLR3 (*Figure 1—figure supplement 1C*).

The expression of cytokines and chemokines in response to S protein was dose dependent (*Figure 1B*). Similarly, S protein-induced inflammatory response was time dependent, with highest being at 8 hr post stimulation (*Figure 1C*). Notably, heat-denatured S2 protein failed to stimulate THP-1 cells, confirming the specificity of S protein and the requirement of its native structural configuration in inducing inflammatory response (*Figure 1—figure supplement 1D*). S1 and S2 proteins from a different source (R&D) were equally potent in inducing proinflammatory cytokines and chemokines (*Figure 1—figure supplement 1E*). Notably, trimeric S protein (S-tri) is also immunostimulatory, producing cytokines *IL6* and *IL1B*, and chemokines *CXCL1* and *CXCL2* in THP1 cells (*Figure 1—figure supplement 1E*). To obtain direct evidence that S protein can induce inflammatory mediators in human immune cells, we stimulated human peripheral blood mononuclear cells (PBMCs) with S2. There was robust induction of *IL1B, IL6, TNFA, CXCL1,* and *CXCL2* in hPBMC at 4 hr post-stimulation (*Figure 1D*).

SARS-CoV-2 does not infect mouse cells since S protein does not bind with mouse ACE2 receptor. To understand whether recognition of S protein by ACE2 is required for induction of inflammatory molecules, we stimulated mouse bone marrow-derived macrophages (mBMDMs) with S1 and S2. Surprisingly, both S1 and S2 proteins triggered expression of *Il6, Il1b, Tnfa, Cxcl1* and *Cxcl2* in mBMDMs (*Figure 1—figure supplement 2A*). Similar to THP1 cells, mBMDMs did not produce type-I and type-II interferons in response to S protein (*Figure 1—figure supplement 2A*). Murine macrophage cell line RAW264.7 cells also responded to S2 protein, producing *Il6, Tnfa,* and *Il1b* (*Figure 1—figure supplement 2B*). To further examine the role of ACE2 in S-induced inflammatory response, we stimulated THP1 cells with S1 or S2 in the presence or absence of ACE2 inhibitor (MLN-4760). Interestingly, inhibition of ACE2 did not affect cytokine production in response to S protein (*Figure 1E*). Taken together, SARS-CoV-2 S protein induces proinflammatory cytokines and chemokines in macrophages and monocytes in an ACE2-independent manner.

## Epithelial cells produce inflammatory mediators in response to SARS-CoV-2 S protein

SARS-CoV-2 primarily infects epithelial cells of the lung, kidney, intestine, and vascular endothelial cells (*Hoffmann et al., 2020*; *Xiao et al., 2020*; *Hamming et al., 2004*; *Varga et al., 2020*). However, it is poorly understood whether SARS-CoV-2-infected epithelial cells produce proinflammatory cytokines and contribute to cytokine storm of COVID-19 patients. To address this concern, we stimulated human lung epithelial cells A549 with S1 or S2 proteins. However, there was no induction of *IL6, IL1B, TNFA, CXCL1,* and *CXCL2* in A549 cells in response to S proteins at 4 hr post-stimulation (*Figure 2—figure supplement 1*). Given that epithelial cells are weaker than innate immune cells in expressing inflammatory mediators, we wondered whether the expression of inflammatory molecules in epithelial cells is delayed. We, therefore, measured cytokines and chemokines in A549 cells following stimulation with S1 and S2 at 12 and 24 hr. Interestingly, both S1 and S2 proteins induced proinflammatory cytokines *IL6, IL1B, TNFA,* and chemokines *CXCL1* and *CXCL2* with highest being at 24 hr post-stimulation (*Figure 2A*). Similarly, Calu3 cells, a commonly used human lung epithelial cell line, and mouse primary lung epithelial cells produced *IL6, IL1B, TNFA, CXCL1,* and *CXCL2* in response to S

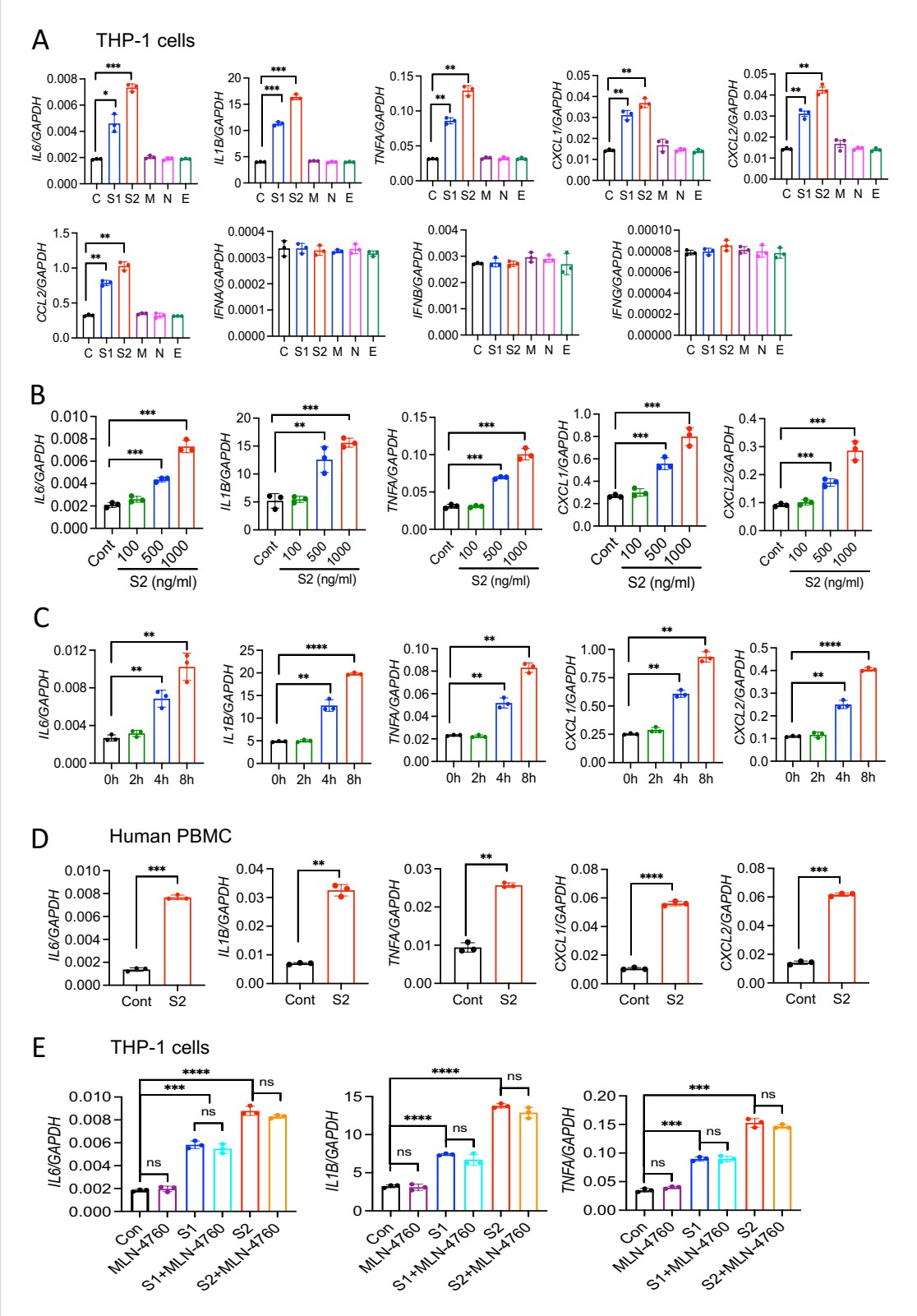

**Figure 1.** SARS-CoV-2 S protein induces cytokines and chemokines in macrophages and monocytes. (**A**) Human monocytic cells THP1-derived macrophages were stimulated with recombinant S1, S2, M, N, and E proteins of SARS-CoV-2 at a concentration of 500 ng/ml. Four hours post-stimulation, the expression of *IL6, IL1B, TNFA, CXCL1, CXCL2, CCL2, IFNA, IFNB,* and *IFNG* was measured by real-time RT-PCR. (**B**) THP1 cells were stimulated with S2 protein at various concentrations for 4 hr and measured the indicated cytokines by real-time RT-PCR. (**C**) THP1 cells were stimulated

*Figure 1 continued on next page*

*Figure 1 continued*

with S2 protein (500 ng/ml). RNA isolated at 2, 4, and 8 hr post-stimulation was measured for *IL6, IL1B, TNFA, CXCL1,* and *CXCL2* by real-time RT-PCR. (**D**) Human peripheral blood mononuclear cells (PBMCs) were incubated with S2 (500 ng/ml) protein for 4 hr. The expression of *IL6, IL1B, TNFA, CXCL1,* and *CXCL2* was measured by real-time RT-PCR. (**E**) THP1 cells were incubated with S1 (500 ng/ml) or S2 (500 ng/ml) in the presence or absence of ACE2 inhibitor MLN-4760 (10 mM). The expression of cytokines was measured at 4 hr by real-time RT-PCR. Data represent mean ± SD (n=3); *p<0.05, **p<0.001, ***p<0.0001, ****p<0.00001 by unpaired Student's t-test. Experiments described in (**A**) were repeated three times, and (**B–E**) were repeated two times. Data of representative experiments are presented.

The online version of this article includes the following source data and figure supplement(s) for figure 1:

**Source data 1.** Raw source data for A-E.

**Figure supplement 1.** SARS-CoV-2 S protein induces inflammatory cytokines in macrophages.

**Figure supplement 1—source data 1.** Raw source data for A-E.

**Figure supplement 2.** Mouse macrophages are stimulated by SARS-CoV-2 S protein.

**Figure supplement 2—source data 1.** Raw source data for A-B.

---

protein (*Figure 2B and C*). We confirmed the expression of IL-1β, IL-6, and TNFα at protein levels in Calu3 and primary lung epithelial cells by ELISA (*Figure 2D and E*).

## Epithelial cells expressing S protein trigger inflammation in macrophages

We next verified whether cytosolic S protein can stimulate epithelial cells to produce inflammatory molecules. Therefore, we transfected A549, Calu3, and HEK293T cells with plasmids expressing S protein or green fluorescent protein (GFP). The expression of S protein in HEK293T, A549, and Calu3 cells was confirmed by Western blotting (*Figure 3—figure supplement 1A*). However, cytosolic S protein did not induce any cytokines and chemokines in A549, Calu3, and HEK293T cells (*Figure 3—figure supplement 1B-D*). These data suggest that cytosolic S protein does not trigger inflammation in epithelial cells.

Since airway and other epithelial cells are primary targets for SARS-CoV-2, we wondered whether virally infected epithelial cells trigger inflammatory response in macrophages and monocytes in a paracrine manner. To address this concern, we collected culture supernatant of A549-S, Calu3-S, or HEK293T-S cells and added (30% V/V) into THP1 cells in culture (*Figure 3A*). However, culture supernatants of epithelial cells expressing S protein failed to induce cytokines *IL6, IL1B,* and *TNFA* in THP1 cells (*Figure 3B and C*, and *Figure 3—figure supplement 2A*). In fact, S protein was not detectable in the culture supernatant of HEK293T and A549 cells expressing S protein (*Figure 3—figure supplement 2B*). Flow cytometric analysis suggested that S protein is primarily located in the cytoplasm, not on the cell surface (*Figure 3—figure supplement 2C*).

We then sought to examine whether innate immune cells become activated when they physically interact with S protein-expressed epithelial cells. Hence, we co-cultured S protein expressing A549, Calu3, or HEK293T cells with THP1 cells at 1:2 ratio (*Figure 3D*). Interestingly, inflammatory cytokines were induced in co-cultured cells (*Figure 3E and F*, and *Figure 3—figure supplement 2D*). We confirmed protein concentrations of IL-6, IL-1β, and TNFα in the culture supernatants of co-cultured cells by ELISA (*Figure 3G and H*, *Figure 3—figure supplement 2E*). Notably, similar to *Figure 3—figure supplement 2B*, S protein was not detected in the culture supernatant of co-cultured cells as measured by ELISA (data not shown), suggesting that THP1 cells are activated by S protein-transfected epithelial cells through an unknown mechanism.

To further confirm that cytosolic S protein expressed in epithelial cells can stimulate macrophages, we lysed HEK293T-S cells, and added the cell lysates into THP1 cells in culture (*Figure 3—figure supplement 2F*). Lysates of HEK293T-S cells efficiently induced inflammatory cytokines in THP1 cells while no such induction was observed in response to cell lysate of HEK293T-GFP cells (*Figure 3—figure supplement 2G*). Taken together, these data imply that SARS-CoV-2 infected epithelial cells may stimulate macrophages and monocytes in a paracrine manner to produce inflammatory mediators.

## S protein activates the NF-κB pathway

Inflammatory genes are transcriptionally regulated by transcription factors that are activated by signaling pathways such as NF-κB, MAPK, STAT3, and AKT. To obtain further insight into how S

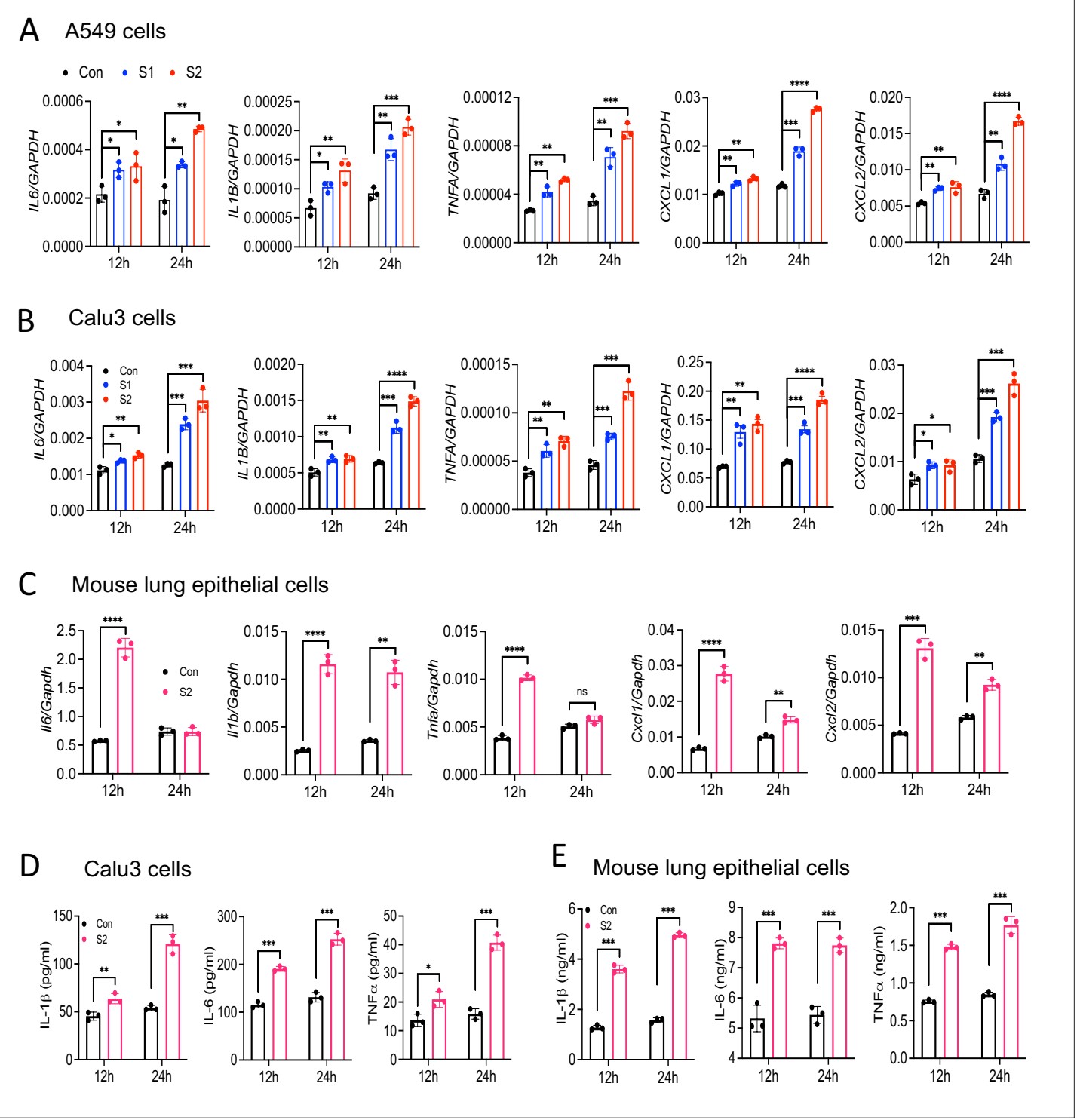

**Figure 2.** Lung epithelial cells produce inflammatory molecules in response to SARS-CoV-2 S protein. (**A, B**) A549 or Calu3 cells were incubated with SARS-CoV-2 S1 or S2 proteins (500 ng/ml) for 12 and 24 hr. The expression of inflammatory cytokines and chemokines was measured by real-time RT-PCR. (**C**) Primary mouse lung epithelial cells were stimulated with S2 (500 ng/ml) for 12 and 24 hr. The expression of inflammatory cytokines and chemokines was measured by real-time RT-PCR. (**D, E**) Calu3 cells or mouse lung primary epithelial cells were stimulated with S2 (500 ng/ml). Culture supernatant collected at 12 and 24 hr were analyzed for IL-6, IL-1β, and TNFα by ELISA. *p<0.05, **p<0.001, ***p<0.0001, ****p<0.00001 by unpaired Student's t-test. Experiments in (**A, B**) were repeated three times. Other experiments were repeated two times and data of representative experiments are presented.

*Figure 2 continued on next page*

*Figure 2 continued*

The online version of this article includes the following source data and figure supplement(s) for figure 2:

**Source data 1.** Raw source data for A-E.

**Figure supplement 1.** Epithelial cells do not respond to SARS-CoV-2 S protein acutely.

**Figure supplement 1—source data 1.** Raw source data.

protein induces the expression of inflammatory mediators, we stimulated THP1 cells with S2 protein. Cell lysates collected at various times following stimulation were analyzed for the activation of these inflammatory pathways by Western blotting. As shown in *Figure 4A*, P65 and IκB were phosphorylated in cells treated with S2. MAPK pathways, including ERK, P38, and JNK, are often activated concomitant to the NF-κB pathway. Surprisingly, there was no activation of ERK and JNK in S2 stimulated cells (*Figure 4A*). There was also no activation of the AKT pathway (*Figure 4A*), but STAT3 was phosphorylated at 2 hr following stimulation (*Figure 4A*). Inflammatory cytokines, such as IL-6, can activate STAT3; thus, the observed activation of STAT3 could be a secondary response of S protein-mediated activation of the NF-κB pathway. S2 protein also activated the NF-κB and STAT3 pathways in A549 cells (*Figure 4B*). To confirm that S protein-induced inflammation was NF-κB dependent, we inhibited the NF-κB pathway using Sc514, an inhibitor of IKKβ, during stimulation with S protein. As expected, inhibition of the NF-κB pathway abrogated inflammatory responses in S protein-stimulated macrophages (*Figure 4C and D*).

## S protein-mediated activation of the NF-κB pathway is TLR2 dependent

Upon recognition of diverse PAMPs at the cell surface or in the endosome, TLRs activate the NF-κB and MAPK pathways through the adapter protein MyD88. To verify if TLR pathways are involved in S protein-mediated activation of the NF-κB pathway, we stimulated WT and *Myd88*$^{-/-}$ BMDM with S2. Interestingly, there was no activation of the NF-κB pathway in *Myd88*$^{-/-}$ BMDM (*Figure 5A*). Consistently, there was no cytokine expression in *Myd88*$^{-/-}$ macrophages upon stimulation with S2 protein (*Figure 5B*). This observation suggests that S protein-mediated activation of the transcription factor NF-κB involves TLR/MyD88 signaling axis. We then interrogated which TLR senses S protein. Since S is a glycoprotein, we anticipated that TLR2, a receptor for lipoprotein, or TLR4, which senses lipopolysaccharide and several other stimuli (*Kawai and Akira, 2006*), could be the immune sensor for S protein. Therefore, we stimulated *Tlr2*$^{-/-}$ and *Tlr4*$^{-/-}$ BMDMs with S2 protein and measured the activation of the NF-κB pathway. There was no activation of the NF-κB pathway in *Tlr2*$^{-/-}$ BMDM, while activation of this pathway was intact in *Tlr4*$^{-/-}$ macrophages (*Figure 5C*). We confirmed that *Tlr2*$^{-/-}$ macrophages were defective in sensing Pam3CSK4, a ligand for TLR2, while they were fully responsive to TLR4 ligand LPS (*Figure 5—figure supplement 1*). Similar to S2, S1 protein activated the NF-κB pathway in WT macrophages but not *Tlr2*$^{-/-}$ macrophages (*Figure 5D*). Consistent to the defective activation of the NF-κB pathway, there was no induction of proinflammatory cytokines in S-stimulated *Tlr2*$^{-/-}$ macrophages (*Figure 5E*). Notably, similar to S protein, S-tri is also sensed by TLR2, but not TLR4, as *Tlr2*$^{-/-}$ macrophages failed to induce cytokines in response to S-tri, while no such defect in inflammatory response was observed in *Tlr4*$^{-/-}$ macrophages (*Figure 5F*). Further, inhibition of TLR2 by C29 blocked S2-mediated induction of proinflammatory cytokines in THP1 cells (*Figure 5G*). C29 also inhibited S-mediated induction of cytokines and chemokines in Calu3 cells (*Figure 5—figure supplement 2*).

Finally, to understand whether S protein induces inflammation in vivo and the role of TLR2 in this process, we administered S1 and S2 proteins into WT or *Tlr2*$^{-/-}$ mice intraperitoneally (i.p.). 16 hr following administration of S protein, we measured the cytokine IL-6, IL-1β, and TNFα in the serum by ELISA. The concentrations of IL-6, IL-1β, and TNFα were elevated following administration of S proteins in WT mice, whereas no such induction was observed in *Tlr2*$^{-/-}$ mice (*Figure 5H*). These data suggest that TLR2 is an innate immune sensor for SARS-CoV-2 S protein, and recognition of S protein by TLR2 leads to the activation of NF-κB and induction of inflammatory mediators.

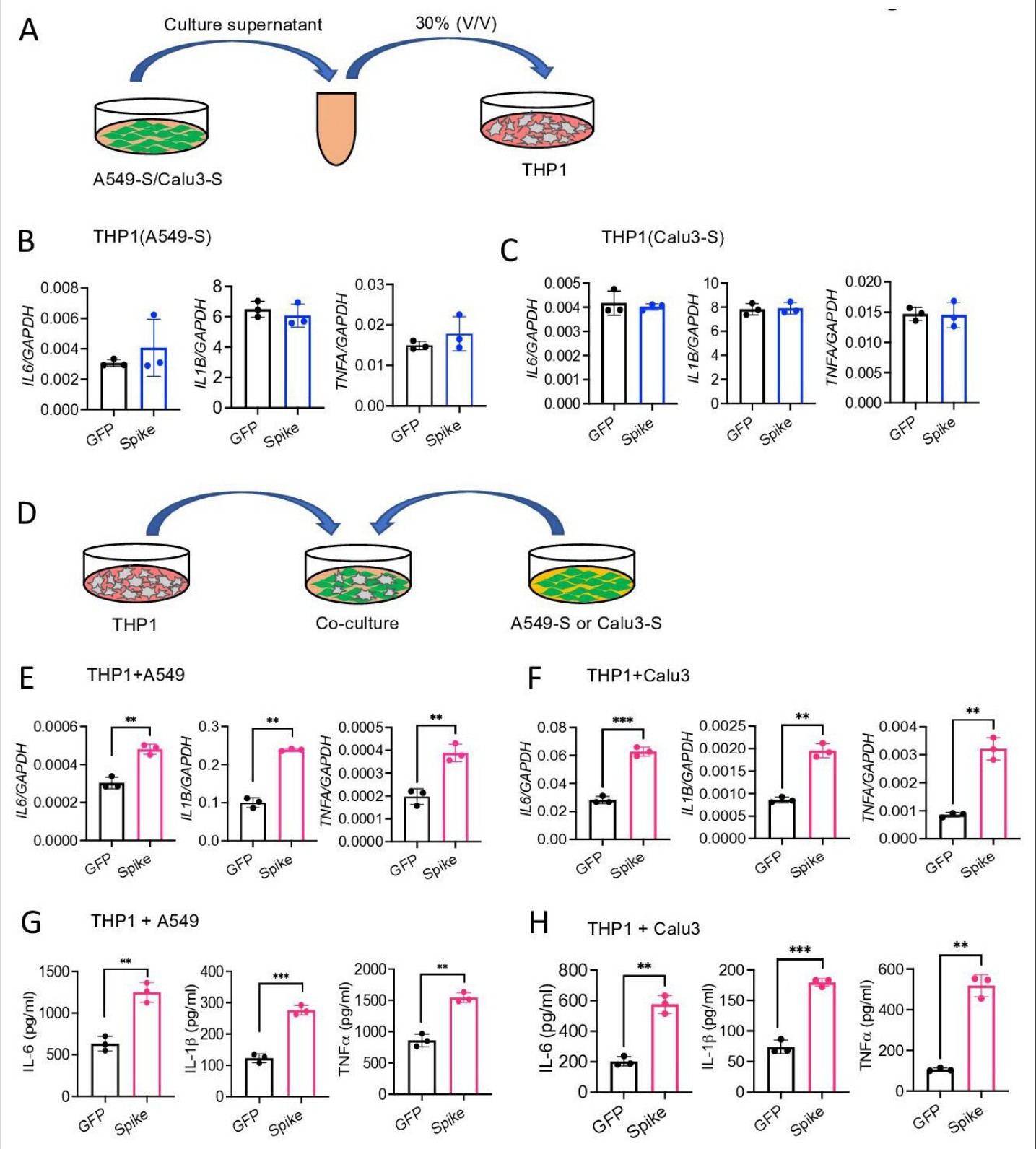

**Figure 3.** Epithelial cells expressing S protein stimulate macrophages during co-culture. (**A–C**) SARS-CoV-2 S protein was overexpressed in A549 or Calu3 cells. Forty-eight hours following transfection with S or GFP plasmids, cell culture supernatants were collected and added into THP1 cells in culture at 30% v/v. (**B, C**) The expression of *IL6*, *IL1B*, and *TNFA* in THP1 cells at 4 hr was measured by real-time RT-PCR. (**D**) A549 or Calu3 cells expressing S protein were co-cultured with THP1 cells at 1:2 ratio for 16 hr. (**E, F**) The expression of *IL6*, *IL1B*, and *TNFA* was measured by real-time

*Figure 3 continued on next page*

*Figure 3 continued*

RT-PCR. (**G, H**) Protein levels of IL-6, IL-1β, and TNFα in culture supernatant described in (**D**) were measured by ELISA. Data represent mean ± SD (n=3); **p<0.001, ***p<0.0001 by unpaired Student's t-test. Experiments were repeated three times and data of representative experiments are presented.

The online version of this article includes the following source data and figure supplement(s) for figure 3:

**Source data 1.** Raw source data for B, C, E, F, G, H.

**Figure supplement 1.** Cytosolic S protein dose not trigger inflammation in epithelial cells.

**Figure supplement 1—source data 1.** Raw source data for B-D.

**Figure supplement 2.** HEK293T cells expressing S protein activate macrophages.

**Figure supplement 2—source data 1.** Raw source data for A, B, D, E, G.

## TLR2 requires heterodimerization with TLR1 or TLR6 for sensing S protein

It is known that TLR2 forms a heterodimer with either TLR1 or TLR6 on the cell surface that promotes ligand binding and propagation of signaling to the downstream kinases (*Buwitt-Beckmann et al., 2006*; *Chang et al., 2007*; *Farhat et al., 2008*; *Jin et al., 2007*). We, therefore, asked whether TLR1 or TLR6 is involved in sensing S-mediated inflammatory response. To this end, we used engineered HEK293T cells which selectively express TLR2 (HEK-Blue-TLR2), TLR2 and TLR1 (HEK-Blue-TLR2/1), or TLR2 and TLR6 (HEK-Blue-TLR2/6) along with SEAP (secreted embryonic alkaline phosphatase), a reporter for NF-κB activation. When we stimulated reporter HEK-Blue cells with S1, S2, or S-tri, all three spike proteins activated NF-κB in HEK-Blue-TLR2, HEK-Blue-TLR1/2, and HEK-Blue-TLR2/6 cells, but not in HEK-Blue-Null and HEK-Blue-TLR4 cells (*Figure 6A and B*). We used Pam3CSK4, FSL1, and LPS as control ligands for HEK-Blue-TLR2/1, HEK-Blue-TLR2/6, and HEK-Blue-TLR4 cells, respectively (*Figure 6A and B*). Notably, HEK293T cells endogenously express TLR1 and TLR6 (*Buwitt-Beckmann et al., 2006*) therefore, HEK-Blue-TLR2 cells responded to both TLR2/1 ligand Pam3CSK4 and TRL2/6 ligand FSL1 (*Figure 6A and B*). We further examined S-induced activation of NF-κB in TLR2, TLR1/2, and TLR2/6 cells by Western blotting. As shown in *Figure 6C*, NF-κB was activated in S2 stimulated HEK-Blue-TLR2, HEK-Blue-TLR1/2, and HEK-Blue-TLR2/6 cells but not in HEK-Blue-Null and HEK-Blue-TLR4 cells. Consistently, the S-induced expression of *IL6* and *IL1B* was similar in HEK-Blue-TLR2, HEK-Blue-TLR2/1, and HEK-Blue-TLR2/6 cells (*Figure 6D*).

Since we observed that both HEK-Blue-TLR1/2 and HEK-Blue-TLR2/6 cells responded to S protein, we wanted to clarify whether TLR1 or TLR6 can sense S protein independent of TLR2. To this end, we inhibited TLR2 in HEK-Blue-TLR cells with C29 during stimulation with S protein. C29 completely blocked S-mediated NF-κB activation in HEK-Blue-TLR2, HEK-Blue-TLR1/2, and HEK-Blue-TLR2/6 cells (*Figure 6E and F*), further confirming the essential role of TLR2 in S-mediated innate immune response. Notably, no activation of the reporter gene was observed in HEK-Blue-TLR2 or HEK-Blue-TLR4 cells stimulated with M, N, and E proteins (*Figure 6—figure supplement 1A, B*), further confirming that S, but not other structural proteins, is primarily responsible for the inflammatory response.

With the above observation, we interpreted that TLR1 and TLR6 plays a redundant role or these co-receptors are dispensable in S-induced inflammatory response. To examine these possibilities, we knocked out *TLR1*, *TLR2*, *TLR6*, or both *TLR1* and *TLR6* in Raw264.7 cells with CRISPR/Cas9 (*Figure 6—figure supplement 2A, B*). On stimulation of null or different TLR knockout cells with S2, TLR1-KO and TLR6-KO cells showed no defect in the S-induced inflammatory responses (*Figure 6G*). Interestingly, TLR1 and TLR6 double KO cells failed to induce any cytokine during stimulation with S protein (*Figure 6G*). Taken together, these data suggest that TLR2 requires dimerization with either TLR1 or TLR6 for sensing S protein and inducing subsequent inflammatory responses.

## Discussion

Both SARS-CoV-2 infection and aberrant host immune responses are responsible for COVID-19 pathogenesis (*Vabret et al., 2020*; *Tay et al., 2020*; *Blanco-Melo et al., 2020*; *Grant et al., 2021*). The initial host immune response against SARS-CoV-2 infection involves innate immune cells, such as macrophages, monocytes, neutrophils, and dendritic cells (*Liao et al., 2020*; *Zhou et al., 2020a*). Cytokines, chemokines, and other inflammatory mediators produced by these cells inhibit virus replication, heal the damage, and activate the adaptive immune system. However, uncontrolled release of cytokines,

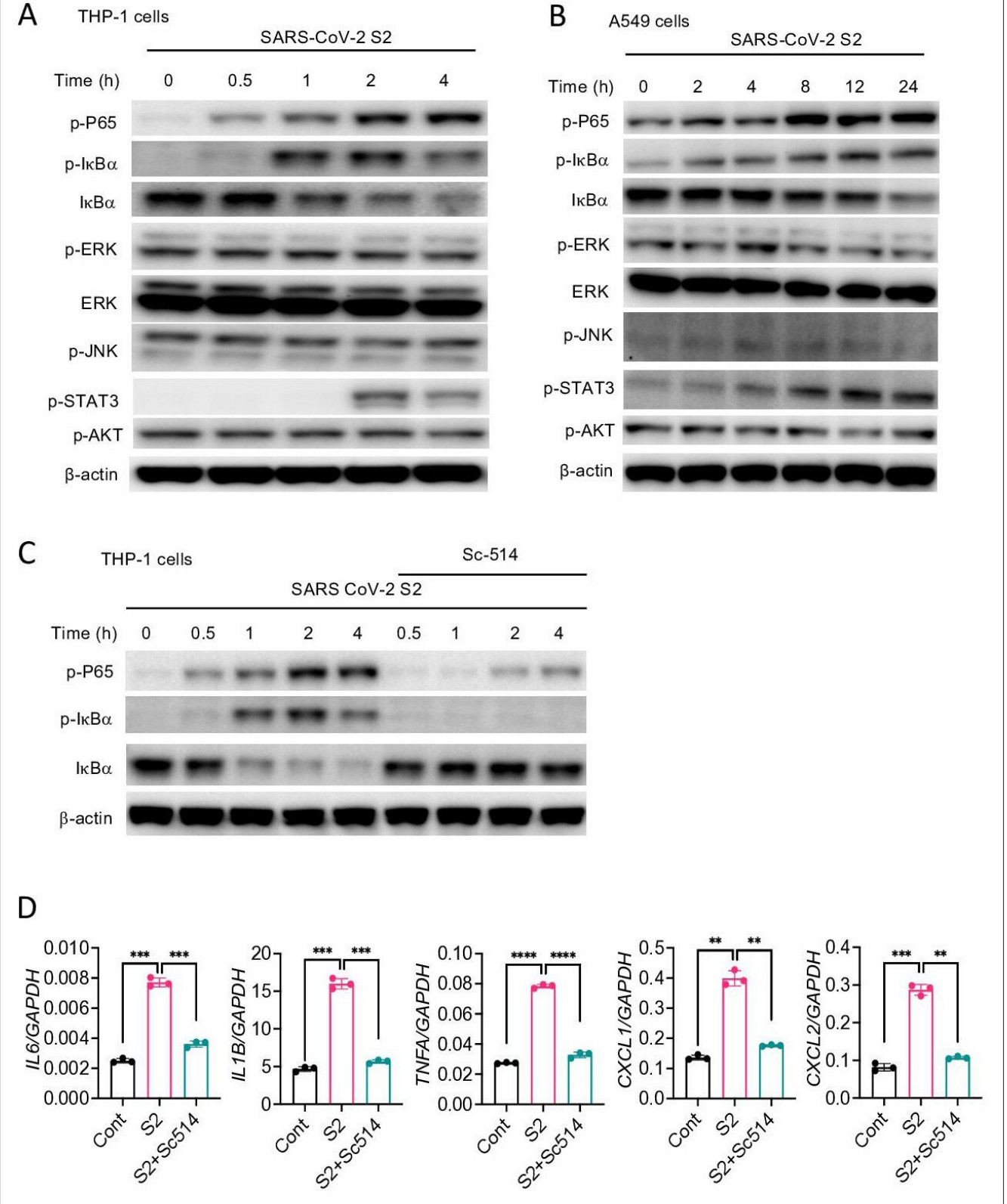

**Figure 4.** SARS-CoV-2 S protein activates the NF-κB pathway. (**A, B**) THP1 and A549 cells were stimulated with S2 (500 ng/ml) for indicated time points. Phosphorylation of P65, IκBα, ERK, JNK, STAT3, and AKT was measured by Western blotting. (**C, D**) THP1 cells were stimulated by SARS-CoV-2 S2 protein (500 ng/ml) in the presence or absence of IKKβ inhibitor sc514. Phosphorylation of P65 and IκBα was measured by Western blotting (**C**). The expression of *IL6, IL1B, TNFA, CXCL1*, and *CXCL2* in stimulated THP1 cells was measured by real-time RT-PCR (**D**). Data represent mean ± SD (n=3);

*Figure 4 continued on next page*

*Figure 4 continued*

*p<0.05, **p<0.001, ***p<0.0001, ****p<0.00001 by unpaired Student's t-test. Experiments in (**A, B**) were repeated three times and (**C, D**) were repeated two times. Data of representative experiments are presented.

The online version of this article includes the following source data for figure 4:

**Source data 1.** Raw source data for D.

chemokines, and reactive oxygen and nitrogen species often exert pathological consequences such as tissue injury, systemic inflammation, and organ failure (*Vabret et al., 2020*; *Tay et al., 2020*; *Blanco-Melo et al., 2020*). Non-surviving COVID-19 patients exhibited a massive influx of macrophages and neutrophils, but reduced T cells in their blood (*Liao et al., 2020*), pointing to the association of hyper-activation of innate immune cells with COVID-19 pathogenesis. Indeed, the innate immune response is heightened in the lung of COVID-19 patients (*Zhou et al., 2020a*; *Xiong et al., 2020*). A better understanding of the mechanism through which SARS-CoV-2 stimulates innate immune cells and activates inflammatory signaling pathways is key to finding better treatment regimens for COVID-19. Our finding that SARS-CoV-2 S protein is a potent viral PAMP involved in the induction of inflammatory cytokines and chemokines via TLR2-dependent activation of the NF-κB pathway, therefore, is a valuable addition to the tremendous scientific effort aiming at combating COVID-19.

Being an RNA virus, SARS-CoV-2 may activate RNA sensors TLR7, RIG-I, and MDA5, which are primarily responsible for production of type I interferons (*Kawasaki and Kawai, 2014*). Interestingly, the type I interferon response is attenuated in COVID-19 patients and SARS-CoV-2 infected cells (*Blanco-Melo et al., 2020*; *Hadjadj et al., 2020*). Transcriptomic analysis of bronchoalveolar lavage fluid and PBMCs of COVID-19 patients also demonstrated higher expression of proinflammatory cytokines and chemokines, but not type I interferons (*Xiong et al., 2020*). Our knowledge of the type I interferon response of SARS-CoV-2 infected macrophages is limited. However, previous studies on SARS-CoV-1, which share 80% similarity with SARS-CoV-2, showed that macrophages and dendritic cells infected by SARS-CoV-1 produce chemokines CXCL10 and CCL2, but not type I interferons (*Cheung et al., 2005*; *Law et al., 2005*). Lack of an interferon response can be explained by the fact that several structural and non-structural proteins including M, N, PLP, ORF3b, ORF6, and NSP1 inhibit type I interferon signaling (*Siu et al., 2014*; *Frieman et al., 2009*; *Narayanan et al., 2008*; *Devaraj et al., 2007*; *Frieman et al., 2007*; *Kopecky-Bromberg et al., 2007*). Despite this evidence, the precise mechanism of excessive production of inflammatory cytokines along with reduced expression of type I interferons in COVID-19 patients remains elusive. In this regard, our findings that SARS-CoV-2 S protein is a potential trigger for proinflammatory cytokines and chemokines help understand why the inflammatory response in COVID-19 is marked by elevated levels of proinflammatory cytokines and chemokines, but poor type I interferon response. Our findings suggest that S protein of SARS-CoV-1 and SARS-CoV-2 shares similar inflammatory function (*Wang et al., 2007*; *Dosch et al., 2009*). Further studies are required to clarify the relative contributions of S protein, viral RNA, and other non-structural proteins in COVID-19-associated cytokine storm.

Inflammatory responses of COVID-19 patients are mostly contributed by innate immune cells, but they weakly express ACE2 (*Ropa et al., 2021*). There is no strong evidence that SARS-CoV-2 infects and propagates infection in immune cells. Thus, it is intriguing how innate immune cells become activated to produce inflammatory mediators during SARS-CoV-2 infection. We propose three mechanisms involved in hyperinflammatory responses during SARS-CoV-2 infection. First, innate immune cells like macrophages and monocytes recognize S protein of SARS-CoV-2 at the cell surface through TLR2, leading to the activation of the NF-κB pathway. Immune sensing of S protein is independent of ACE2 since mouse macrophages, whose ACE2 receptor does not bind to S protein, express inflammatory cytokines and chemokines in response to S protein. Further, inhibition of ACE2 does not impair S-induced inflammatory response, indicating that immunostimulatory function of S protein is independent of ACE2. Second, innate immune cells get activated by virally infected epithelial cells. Our data suggest that epithelial cells expressing S protein in the cytosol can activate macrophages when they physically interact. Although the underlying mechanism is not clear, macrophages may engulf or recognize cell surface molecules expressed on SARS-CoV-2 infected epithelial cells. In a third mechanism, like myeloid cells, epithelial cells can be activated by S protein extracellularly, leading to the induction of proinflammatory cytokines and chemokines. Although inflammatory responses of epithelial cells are weaker than that of innate immune cells, epithelial cell-derived chemokines recruit

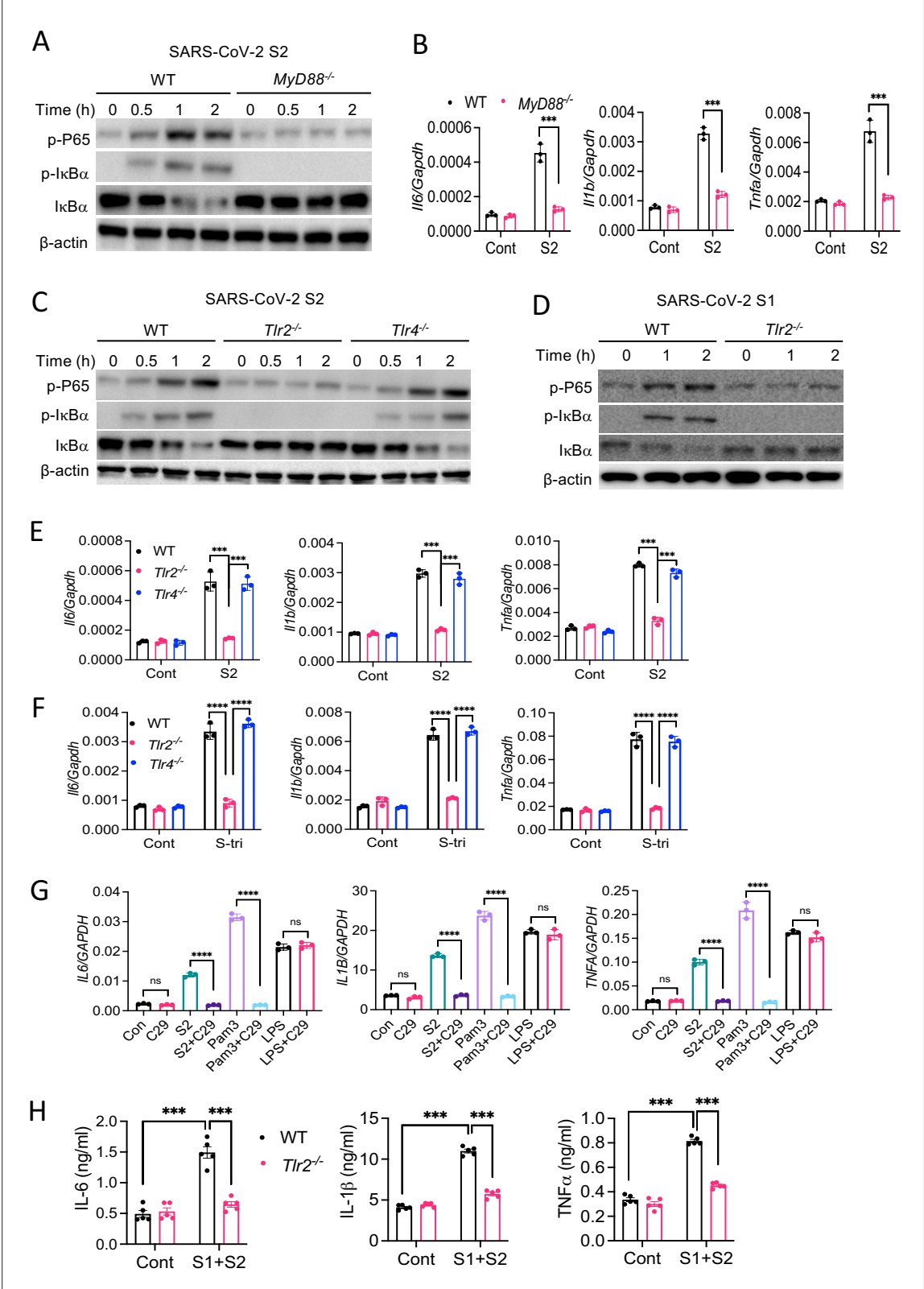

**Figure 5.** TLR2 recognizes SARS-CoV-2 S protein and activate the NF- κ B pathway. (**A, B**) Bone marrow-derived macrophages (BMDMs) from WT and *Myd88⁻/⁻* mice were stimulated with S2 protein (500 ng/ml). (**A**) The activation of the NF- κ B pathway was measured by Western blot analysis of P-P65 and P-I κ Bα. (**B**) The induction of *Il6, Il1b,* and *Tnfa* was measured by real-time RT-PCR. (**C**) BMDMs from WT, *Tlr2⁻/⁻*, and *Tlr4⁻/⁻* mice were treated with S2 protein (500 ng/ml). Cell lysates collected at different times were analyzed for the activation of the NF- κ B pathway by Western blotting of P-P65

*Figure 5 continued on next page*

*Figure 5 continued*

and P-I $\kappa$ B$\alpha$. (**D**) BMDMs from WT and *Tlr2*$^{-/-}$ mice were treated with S1 protein (500 ng/ml), and the activation of P65 and I$\kappa$B$\alpha$ was measured by Western blotting. (**E, F**) WT, *Tlr2*$^{-/-}$, and *Tlr4*$^{-/-}$ macrophages were treated with S2 protein (500 ng/ml) or S-tri (500 ng/ml). The expression of cytokines was measured by real-time RT-PCR at 4 hr post-stimulation. Data represent mean ± SD (n=3); ***p<0.0001, ****p<0.00001 by unpaired Student's t-test. Experiments were repeated two times and data of representative experiments are presented. (**G**) THP1 cells were stimulated with S2 protein (500 ng/ml), Pam3CSK4 (500 ng/ml), or LPS (100 ng/ml) in the presence or absence of Tlr2 inhibitor C29 (150 mM) for 4 hr. The expression of *IL6, IL1B*, and *TNFA* was measured by real-time RT-PCR. (**H**) WT and *Tlr2*$^{-/-}$ mice were administered with S1 and S2 protein (1 µg each/mouse). Blood collected before and 16 hr post S protein administration was measured for IL-6, IL-1β, and TNFα by ELISA. Data represent mean ± SEM (n=5); ***p<0.0001, ****p<0.00001 by unpaired Student's t-test. Experiments were repeated two times and data of representative experiments are presented.

The online version of this article includes the following source data and figure supplement(s) for figure 5:

**Source data 1.** Raw source data for B, E, F, G, H.

**Figure supplement 1.** Macrophages of *Tlr2*$^{-/-}$ mice are defective in sensing TLR2 ligand Pam3CSK4.

**Figure supplement 1—source data 1.** Raw source data.

**Figure supplement 2.** Inhibition of TLR2 abrogates S-mediated inflammatory response in Calu-3 cells.

**Figure supplement 2—source data 1.** Raw source data.

neutrophils, monocytes, and lymphocytes in SARS-CoV-2-infected lungs and thereby contribute to immunopathology of COVID-19 patients.

Our data demonstrate that TLR2 is the innate immune sensor for the S protein. Being a sensor for lipopeptides and lipoproteins, TLR2 plays a critical role in host defense against many bacterial and viral infections (*Oliveira-Nascimento et al., 2012*). Unlike other TLRs, ligand recognition by TLR2 involves dimerization with TLR1 or TLR6 (*Buwitt-Beckmann et al., 2006*; *Jin et al., 2007*). Since an absence of either TLR1 or TLR6 does not impair S-induced inflammatory response, it seems that TLR2/1 and TLR2/6 heterodimers play redundant roles in sensing S protein. However, S protein failed to induce inflammatory response in macrophages defective in either TLR2 or both TLR1 and TLR6, suggesting that TLR2 alone cannot sense S protein, but requires dimerization with either TLR1 or TLR6. Although TLR2/1 and TLR2/6 heterodimers are considered receptors for tri-acylated and di-acylated lipopeptides, respectively, it is not uncommon that both TLR1 and TLR6 co-receptors can sense a particular PAMPs. For example, both TLR2/1 and TLR2/6 heterodimers are involved in the inflammatory response to core and NS3 proteins of hepatitis C virus (*Chang et al., 2007*).

Like other MyD88-dependent TLR pathways, ligation of TLR2 leads to the activation of transcription factors NF-κB and AP-1 (*Kawasaki and Kawai, 2014*). Interestingly, while there was activation of NF-κB, AP-1 upstream signaling kinases such as ERK and JNK were not seen activated by S proteins. SARS-CoV-1 S protein activates the NF-κB pathway in human monocyte-derived macrophages (*Wang et al., 2007*; *Dosch et al., 2009*). COVID-19 patients also exhibited increased activation of the NF-κB pathway (*Hadjadj et al., 2020*). Interestingly, in contrast to these findings, a separate study reported that SARS-CoV-1 S protein-expressing baculovirus activates AP-1 but not NF-κB in A549 cells (*Chang et al., 2004*). Future studies dissecting the signaling pathway regulated by S protein of SARS-CoV-1 and CoV-2 may reveal further insights into the mechanisms of S protein-induced inflammation.

Given the importance of understanding the mechanism of hyperinflammatory response during COVID19, other research groups are also focused in exploring SARS-CoV-2 ligands and pathways involved in inflammatory response (*Shirato and Kizaki, 2021*; *Zhao et al., 2021*; *Zheng et al., 2021*). The finding of two recent studies showing that S protein triggers inflammatory response is in agreement with our study (*Shirato and Kizaki, 2021*; *Zhao et al., 2021*). But, unlike our data, these studies showed that S protein is sensed by TLR4 instead of TLR2. However, a separate study demonstrated that SARS-CoV-2 induces inflammation via TLR2, but not TLR4, supporting our data (*Zheng et al., 2021*). Interestingly, it was shown that E protein, but not S protein, is immunostimulatory (*Zheng et al., 2021*). While these studies greatly contributed to our understanding of the mechanism of COVID19 pathogenesis, some of these findings are inconsistent and conflicting. It appears that the recombinant S and E proteins used in those studies were generated in *Escherichia coli*. Thus, it is possible those recombinant proteins were contaminated by bacterial PAMPs. We used recombinant proteins produced in mammalian cell HEK293T, ensuring that the proteins are not contaminated by bacterial PAMPs. Using the endotoxin assay, we further confirmed that the recombinant S proteins are endotoxin-free.

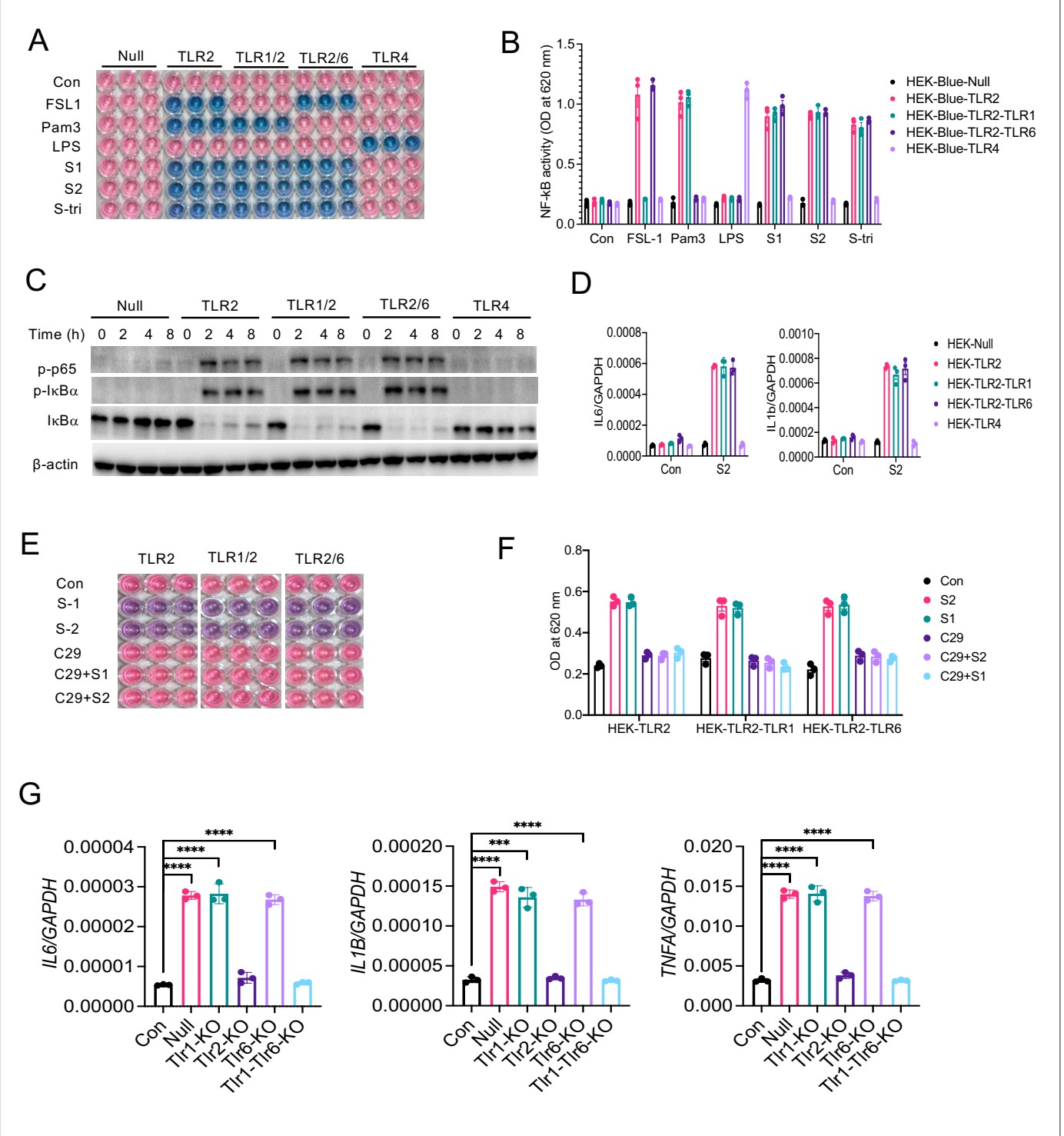

**Figure 6.** TLR1 and TLR6 are dispensable in S-mediated activation of TLR2/NF-κB pathway. (**A, B**) HEK-Blue-Null, HEK-Blue-TLR2, HEK-Blue-TLR1/2, HEK-Blue-TLR2/6, and HEK-Blue-TLR4 were stimulated with S1, S2, or S-tri for 6 hr. FSL1, Pam3CSK4, and LPS were used as ligands for TLR2/1, TLR2/6, and TLR4, respectively. The activation of NF-κB was monitored by the blue color development (**A**), which was measured at 620 nm (**B**). (**C**) HEK-Blue-Null, HEK-Blue-TLR2, HEK-Blue-TLR1/2, HEK-Blue-TLR2/6, and HEK-Blue-TLR4 cells were stimulated with S2 (500 ng/ml) at indicated times. The activation of P-P65 and P-IκBα was measured by Western blot analysis. (**D**) HEK-Blue-Null, HEK2-Blue-TLR2, HEK-Blue-TLR1/2, HEK-Blue-TLR2/6, and HEK-Blue-TLR4 cells were stimulated with S2 (500 ng/ml) for 6 hr. The induction of *IL6* and *IL1B* was measured by real-time RT-PCR. (**E, F**) HEK-Blue-TLR2, HEK-Blue-TLR2/1, and HEK-Blue-TLR2/6 cells were stimulated with S1 or S2 in the presence or absence of TLR2 inhibitor C29 (150 mM) for 6 hr. The NF-κB activity was monitored colorimetrically at 620 nm. (**G, H**) TLR1, TLR2, TLR6, or TLR1/6 were knocked out in Raw264.7 cells with CRISPR/Cas9.

*Figure 6 continued on next page*

*Figure 6 continued*

Cells were then stimulated with S2 protein (500 ng/ml) for 4 hr. (**G**) The expression of cytokines was measured by real-time RT-PCR. Data represent mean ± SD (n=5); ***p<0.0001, ****p<0.00001 by unpaired Student's t-test. All experiments were repeated three times and data of representative experiments are presented.

The online version of this article includes the following source data and figure supplement(s) for figure 6:

**Source data 1.** Raw source data for B, D, F, G.

**Figure supplement 1.** M, N, and E proteins do not activate TLR2 pathway.

**Figure supplement 1—source data 1.** Raw source data.

**Figure supplement 2.** Knocking out of TLRs in Raw264.7 cells with CRISPR/Cas9.

**Figure supplement 2—source data 1.** Raw source data for A-B.

In summary, this study documents a potential mechanism for the inflammatory response induced by SARS-CoV-2. We demonstrate that SARS-CoV-2 S protein is a potent viral PAMP that upon sensing by TLR2 activates the NF-κB pathway, leading to the expression of inflammatory mediators in innate immune and epithelial cells. The effort so far in combating the COVID-19 pandemic is unprecedented, making it possible for the development of a number of vaccines within a year of outbreak. Since S protein is being targeted by most of the vaccine candidates, it is important to consider its inflammatory function in vaccine design. Moreover, it is critically important to investigate whether *TLR2* polymorphisms are associated with COVID19 pathogenesis and the role of TLR2 in antibody production against SARS-CoV-2 vaccines. Considering the fact that new variants of SARS-CoV-2 with mutations in the S protein spread more easily and may confer more severe disease, the effectiveness of current vaccines remain uncertain (*Plante et al., 2021*). Thus, the importance of developing therapeutic drugs for COVID-19 remains high. This study suggests that TLR2 or its downstream signaling adapters could be therapeutically targeted to mitigate hyperinflammatory response in COVID-19 patients.

## Materials and methods
### Mice
C57BL6/J (WT), *Myd88*$^{-/-}$, *Tlr2*$^{-/-}$, and *Tlr4*$^{-/-}$ mice (all C57BL6/J strain), purchased from Jackson Laboratory were used in this study. All mice were bred and maintained in a specific pathogen-free (SPF) facility at the UT Southwestern Medical Center. All studies were approved by the Institutional Animal Care and Use Committee (IACUC) and were conducted in accordance with the IACUC guidelines and the National Institutes of Health Guide for the Care and Use of Laboratory Animals. All experimental groups were conducted with age and sex-matched male and female mice. No masking was used during data collection and analysis.

### Cell culture and maintenance
The human embryonic kidney epithelial cell line HEK293T (ATCC, CRL-3216), and human lung epithelial cell line A549 (ATCC, CCL-185) were cultured in Dulbecco's modified Eagle's medium (DMEM; high glucose, Sigma-Aldrich) supplemented with 10% (v/v) fetal bovine serum (FBS) (Sigma-Aldrich) and 1% (v/v) PenStrep (Sigma-Aldrich) and maintained in a 5% $CO_2$ incubator at 37°C. THP1 (ATCC, CRL-TIB-202) cells were cultured in Roswell Park Memorial Institute (RPMI)–1640 medium (R8758, Sigma-Aldrich) supplemented with 10% (v/v) FBS (Sigma-Aldrich) and 1% (v/v) PenStrep (Sigma-Aldrich) and maintained in a 5% $CO_2$ incubator at 37°C. Calu3 cells (ATCC, HTB-55) were cultured in Eagle's minimum essential medium (302003, ATCC) supplemented with 10% (v/v) FBS (Sigma-Aldrich) and 1% (v/v) PenStrep (Sigma-Aldrich) and maintained in a 5% $CO_2$ incubator at 37°C. All cell lines were authenticated using STR profiling and confirmed to be free from mycoplasma contamination by testing with a mycoplasma detection kit (Sigma-Aldrich).

### In vitro studies with THP-1 macrophage-like cells
Suspended THP-1 cells were cultured in medium containing 100 ng/ml phorbol-12-myristate 13-acetate (PMA; tlrl, Invivogen) to make them adherent macrophage-like cells. Following 24 hr post PMA treatment, THP-1 macrophage-like cells were washed with pre-warmed RPMI-1640 medium containing 10% FBS and 1% penicillin-streptomycin and allowed to grow in PMA-free culture medium for the

next 12 hr. To examine the effect of SARS-CoV-2 structural proteins on inflammatory responses, THP-1 macrophage-like cells were stimulated with SARS-CoV-2 S1 (RayBiotech, 230-30161), SARS-CoV-2 S2 (RayBiotech, 230-30163), SARS-CoV-2 N (RayBiotech, 230-30164), SARS-CoV-2 M (MyBioSource, MBS8574735), and SARS-CoV-2 E (MyBioSource, MBS9141944) for 4 hr. To validate the immune response of S1 and S2 proteins of RayBiotech, we used recombinant S1 (10569-CV-100), S2 (10594-CV-100), and Spike-trimer (10549-CV-100) proteins purchased from R&D Biosystems.

## Culture of mouse bone marrow-derived macrophage

Mouse bone marrow cells were collected as described previously (*Udden et al., 2017*). Bone marrow cells were cultured in L929 cell-conditioned IMDM medium supplemented with 10% FBS, 1% nonessential amino acids, and 1% penicillin-streptomycin for 6 days to differentiate into macrophages. BMDMs were seeded in 12-well cell culture plates at a concentration of $1.2 \times 10^6$ cells/well and incubated overnight before in vitro studies. BMDMs were stimulated as described above.

## Culture and maintenance of HEK-Blue cells

HEK-Blue-Null2 cells (hkb-null2, InvivoGen), HEK-Blue hTLR2 cells (hkb-htlr2, InvivoGen), HEK-Blue-hTLR2-TLR1 cells (hkb-htlr21, InvivoGen), HEK-Blue hTLR2-TLR6 cells (hkb-htlr26, InvivoGen), and HEK-Blue hTLR4 cells (hkb-htlr4, InvivoGen) were cultured and maintained as per the manufacturer's instruction.

## Culture of mouse primary lung epithelial cells

Primary lung epithelial cells were cultured as described earlier (*Kasinski and Slack, 2013*). Briefly, the mouse was sacrificed under aseptic conditions and the lung was perfused with PBS. The lung was dissected out, washed two times with RPMI-1640, and minced with sterile scissors. RPMI-1640 supplemented with collagenase (1 mg/ml) was added to the minced lung and incubated for 1 hr at 37°C in a 5% $CO_2$ incubator. The lung homogenate was mix properly and kept on ice to settle down large pieces of lung tissue. The top-half of the suspension was transferred and centrifuged at 1000×$g$ for 5 min at 4°C. Following removal of supernatant, cell pellets containing lung epithelial cells were resuspend into the RPMI-1640 medium supplemented with 10% FBS and 1% penicillin-streptomycin and cultured on collagen-coated plates at 37°C in a 5% $CO_2$ incubator.

## cDNA constructs and transient transfection

At 50–60% confluency, HEK293T, A549, and Calu3 cells were transfected with GFP-Flag (VB200507-2985cmv) or SARS-CoV-2 S-Flag (VB200507-2984jyv) (1.5 µg/ml) constructs using Lipofectamine 3000 reagent (Invitrogen) according to the manufacturer's instructions, and confirmed by observing GFP under fluorescence microscope and Western blot analysis of SARS-CoV-2 S and Flag proteins. Forty-eight hours post-transfection, cells were lysed with RIPA lysis buffer containing complete protease inhibitor cocktail and phosphatase inhibitor cocktail (Roche) for the detection of S protein by Western blot or ELISA, or resuspended in TRIzol reagent (Invitrogen) for the isolation of RNA and subsequent measurement of S mRNA by real-time RT-PCR.

## CRISPR knockout of TLR in mouse macrophage

Mouse macrophage cells RAW 264.7 (TIB-71, ATCC) were transfected with GFP-Flag or TLR1 CRISPR/Cas9 (sc-423418, Santa Cruz) or TLR2 CRISPR/Cas9 (sc-423981, Santa Cruz), or TLR6 CRISPR/Cas9 (sc-423420, Santa Cruz) constructs (1.5 µg/ml) using Lipofectamine 3000 reagent (Invitrogen) according to the manufacturer's instructions. Transfection was confirmed by observing GFP under fluorescence microscope, whereas, knockdown of respective TLRs was confirmed by real-time RT-PCR.

## In-vitro stimulation of epithelial cells

To examine the effect of SARS-CoV-2 proteins on inflammatory responses in epithelial cells, HEK293T, A549, and Calu3 cells were stimulated with SARS-CoV-2 S1 (RayBiotech, 230-30161), SARS-CoV-2 S2 (Ray Biotech, 230-30163) for 4, 12 or 24 hr. RNA was isolated and measured for the expression of inflammatory genes by real-time PCR.

## Co-culture of macrophages and epithelial cells

HEK293T-GFP and HEK293T-S or A549-GFP and A549-S or Calu3-FGP and Calu3-S cells were cultured with THP1 macrophage-like cells in a ratio of 1:2 (macrophages were twice in number to epithelial cells). Following 16 hr of co-culture, culture medium was collected, filtered with 0.2 µM filter, and used for ELISA. RNA was isolated and measured for the expression of inflammatory genes by real-time RT-PCR.

## Stimulation of macrophages with conditioned medium of S protein expressed epithelial cells

At 50–60% confluency, HEK293T, A549, and Calu3 cells were transfected with GFP-Flag or SARS-CoV-2 S-Flag (1.5 µg/ml) constructs using Lipofectamine 3000 reagent (Invitrogen) according to the manufacturer's instructions, and confirmed by observing GFP under fluorescence microscope and Western blot analysis of SARS-CoV-2 S and Flag. Forty-eight hours post-transfection, culture medium (conditioned medium) was collected, filtered with 0.2 µM filter, and stored at –80°C. At about 85% confluency, culture medium of THP1 cells was replaced with new media containing 30% (v/v) of epithelial cell-conditioned medium. After 4 hr incubation with conditioned medium, the expression of inflammatory cytokines and chemokines were measured.

## Real-time reverse transcription (RT)-PCR

Epithelial cells, BMDMs, THP1 cells, and RAW264.7 cells were lysed in TRIzol reagent (Invitrogen). Total RNA was isolated using TRIzol reagent (Invitrogen) following the manufacturer's instructions. Isolated RNA was reverse transcribed into cDNA using iScript (Bio-Rad). Real-time RT-PCR was performed using iTaq Universal SYBR Green Supermix (Bio-Rad). Individual expression data were normalized to GAPDH as described earlier (*Hu et al., 2015*).

## In-vitro studies with human peripheral blood mononuclear cells

Human PBMCs were obtained from STEMCELL TECHNOLOGIES (70025), and were cultured in RPMI-1640 medium (R8758, Sigma-Aldrich) supplemented with 10% (v/v) FBS (Sigma-Aldrich) and 1% (v/v) penicillin-streptomycin (Sigma-Aldrich) and maintained in a 5% $CO_2$ incubator at 37°C. After 48 hr, hPBMCs were centrifuged and resuspended in fresh medium for 3 hr, and then stimulated with SARS-CoV-2 S (500 ng/mL) for 4 hr.

## Inflammatory response of SARS-CoV-2 S protein in mice

WT and $Tlr2^{-/-}$ mice were intraperitoneally injected with S1 and S2 subunits of SARS-CoV-2 S protein at equal concentration (1 µg each protein/mouse). Blood was collected before S protein administration by cheek puncturing. Sixteen hours following treatment, mice were sacrificed and blood was drawn from the heart. Serum was separated from the blood by centrifugation and used for the measurement of cytokines by ELISA.

## ELISA

BMDMs and THP1 cells were lysed in ice-cold RIPA buffer supplemented with complete protease inhibitor and phosphatase inhibitor cocktails (Roche). Protein concentration was measured by the Pierce BCA Protein Assay Kit (Thermo Fisher Scientific-23227). Serum was isolated from mouse blood by centrifugation at 10,000 RPM for 10 min at 4°C. The concentration of IL-6, IL-1β, and TNF-α in cell culture medium and serum was measured using commercially available ELISA kits (R&D Systems). SARS-CoV-2 S protein in cell lysates was detected by SARS-CoV-2 S ELISA Kit and following the manufacturer's instruction (RayBiotech, ELV-COVID19S2).

## Western blotting

THP1 cells, BMDM, HEK293T, A549, and Calu3 cells were lysed in ice-cold RIPA lysis buffer containing complete protease inhibitor and phosphatase inhibitor cocktails (Roche), resolved by SDS-PAGE, and transferred onto a PVDF membrane. The membranes were immunoblotted with antibodies against Phospho-NF-κB p65 (3033, Cell Signaling), Phospho-IκBα (9246, Cell Signaling), IκBα (4812, Cell Signaling), Phospho-ERK (4370, Cell Signaling), ERK (4695, Cell Signaling), Phospho-JNK (4668, Cell Signaling), Phospho-AKT (4060, Cell Signaling), Phospho-STAT3 (9145, Cell Signaling), SARS-CoV-2

S (GTX632604, GeneTex), and β-actin (A2228, Sigma-Aldrich). Immunoreactive protein bands were detected using ECL super signal west femto substrate reagent (Thermo Fisher Scientific).

## Flow cytometric analysis of SARS-CoV-2 overexpressed HEK293T cells

HEK293T cells were transfected with SARS-CoV-2 S plasmid. Forty-eight hours following transfection, cells were trypsinized and processed for cell surface staining. Briefly, $0.5 \times 10^6$ cells were resuspended in staining buffer (00-4222-26; eBioscience) and centrifuged at 1500 RPM for 3 min at 4°C. Cells were then incubated with Fc block CD16/32 monoclonal antibody (14-0161-82, eBioscience) and stained with primary antibody SARS-CoV-2 (GTX632604, GeneTex) for 30 min on ice. After washing with staining buffer once, cells were stained with secondary antibody DyLight 488 (35552, Thermo Fisher Scientific) on ice for 1 hr. Finally, cells were washed two times with staining buffer and acquired in flow cytometer (CytoFLEX-Beckman Coulter). Flow cytometric data were analyzed by FlowJo software.

## Heat inactivation of S2 protein

S2 protein was heated at 95°C for 30 in. Native of heat-denatured S2 proteins (500 ng/ml) were used to stimulate THP1 cells and measurement of cytokines by real-time RT-PCR.

## In vitro studies with TLR2 and ACE2 inhibitors

THP1 cells were pretreated with C29 (150 µM), a TLR2 inhibitor, for 1 hr. Control and C29-treated cells were then stimulated with SARS-CoV-2 S2 (500 ng/ml), Pam3CSK4 (500 ng/ml), or LPS (100 ng/ml) in the presence or absence of Tlr2 inhibitor C29 (150 µM) for 4 hr. Similarly, Calu3 cells were pretreated with C29 (150 µM) for 1 hr, and then control and C29-treated Calu3 cells were stimulated with SARS-CoV-2 S2 (500 ng/ml) for 24 hr. For, inhibition of ACE2, THP1 cells were treated with MLN-4760 at 10 µM concentration during stimulation with S protein. The expression of inflammatory cytokines was measured by real-time RT-PCR.

## Statistical analysis

Data are represented as mean ± SD or SEM. Data were analyzed by Prism8 (GraphPad Software) and statistical significance was determined by two-tailed unpaired Student's t-test. $p < 0.05$ was considered statistically significant.

# Acknowledgements

The authors would like to thank the UT Southwestern Animal Resource Center (ARC) for maintenance and care of our mouse colony. The authors thank Dr. Zhijian 'James' Chen for sharing Myd88[-/-] mice and Dr. Esra Akbay for sharing A549 cells. Hasan Zaki is supported by The National Institute of Diabetes and Digestive and Kidney Diseases (NIDDK) of the National Institute of Health (NIH) under Award Number R01DK125352, Cancer Prevention and Research Institute of Texas (CPRIT) Individual Investigator Award (RP200284), and American Cancer Society (ACS) Research Scholar Award (RSG-21-021-01-TBE). Rashmin C Savani holds the William Buchanan Chair in Pediatrics and is funded by a Sponsored Research Agreement with Mallinckrodt Pharmaceuticals, Inc for an unrelated project. John W Schoggins is supported by NIH grant AI158254.

# Additional information

## Competing interests

John W Schoggins: Reviewing editor, eLife. The other authors declare that no competing interests exist.

## Funding

| Funder | Grant reference number | Author |
|---|---|---|
| Cancer Prevention and Research Institute of Texas | RP200284 | Hasan Zaki |

| Funder | Grant reference number | Author |
| --- | --- | --- |
| National Institute of Diabetes and Digestive and Kidney Diseases | R01DK125352 | Hasan Zaki |
| National Institutes of Health | AI158124 | John W Schoggins |
| American Cancer Society | Research Scholar RSG-21-021-01-TBE | Hasan Zaki |

The funders had no role in study design, data collection and interpretation, or the decision to submit the work for publication.

## Author contributions
Shahanshah Khan, Data curation, Formal analysis, Investigation, wrote methods and figure legends; Mahnoush S Shafiei, Christopher Longoria, Investigation; John W Schoggins, Writing – review and editing; Rashmin C Savani, Resources, Writing – review and editing; Hasan Zaki, Conceptualization, Data curation, Formal analysis, Funding acquisition, Methodology, Project administration, Resources, Supervision, Writing - original draft, Writing – review and editing

## Author ORCIDs
Shahanshah Khan (iD) http://orcid.org/0000-0003-3052-932X
John W Schoggins (iD) http://orcid.org/0000-0002-7944-6800
Hasan Zaki (iD) http://orcid.org/0000-0001-9002-5399

## Ethics
All studies were approved by the Institutional Animal Care and Use Committee (IACUC) and were conducted in accordance with the IACUC guidelines and the National Institutes of Health Guide for the Care and Use of Laboratory Animals. The IACUC permit number is 2016-101683.

## Decision letter and Author response
Decision letter https://doi.org/10.7554/eLife.68563.sa1
Author response https://doi.org/10.7554/eLife.68563.sa2

## Additional files

### Supplementary files
• Transparent reporting form

### Data availability
There is no clinical data and large data set in this paper. The raw data for all graphs presented in this paper are included as source data.

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

# Appendix 1

## Appendix 1—key resources table

| Reagent type (species) or resource | Designation | Source or reference | Identifiers | Additional information |
|---|---|---|---|---|
| Genetic reagent (*Mus musculus*) | C57BL/6J | Jackson Laboratory | RRID:MGI:3028467 JAX:000664 | |
| Genetic reagent (*M. musculus*) | *Myd88$^{-/-}$* C57BL/6J | Jackson Laboratory | | |
| Genetic reagent (*M. musculus*) | *Tlr2$^{-/-}$* C57BL/6J | Jackson Laboratory | B6.129-Tlr2tm1kir/J Stock No: 004650 | |
| Genetic reagent (*M. musculus*) | *Tlr4$^{-/-}$* C57BL/6J | Jackson Laboratory | B6.B10ScN-Tlr4lps-del/JthJ Stock No: 007227 | |
| Cell line (*Homo sapiens*) | HEK-293T | ATCC | Cat# CRL-3216 | |
| Cell line (*H. sapiens*) | A549 | ATCC | Cat# CCL-185 | |
| Cell line (*H. sapiens*) | THP1 | ATCC | Cat# CRL-TIB-202 | Cat# CRL-TIB-202 |
| Cell line (*H. sapiens*) | Calu-3 | ATCC | Cat# HTB-55 | |
| Cell line (*H. sapiens*) | HEK-Blue-Null2 | InvivoGen | Cat# hkb-null2 | |
| Cell line (*H. sapiens*) | HEK-Blue-hTLR2 | InvivoGen | Cat# hkb-htlr2 | |
| Cell line (*H. sapiens*) | HEK-Blue-hTLR2-TLR1 | InvivoGen | Cat# hkb-htlr21 | |
| Cell line (*H. sapiens*) | HEK-Blue hTLR2-TLR6 | InvivoGen | Cat# hkb-htlr26 | |
| Cell line (*H. sapiens*) | HEK-Blue-hTLR4 | InvivoGen | Cat# hkb-htlr4 | |
| Cell line (*H. sapiens*) | Peripheral blood mononuclear cells | StemCell Technologies | Cat# 70025 | |
| Cell line (*M. musculus*) | Bone marrow-derived macrophage | This paper | | See 'Culture of mouse bone-marrow-derived macrophages' in Materials and methods |
| Cell line (*M. musculus*) | Primary lung epithelial cells | This paper | | See 'Culture of mouse primary lung epithelial cells' in Materials and methods |
| Cell line (*M. musculus*) | RAW264.7 | ATCC | Cat# TIB-71 | |
| Antibody | (Rabbit monoclonal) Phospho-NF-kB p65 | Cell Signaling | Cat# 3033 | WB (1:1000) |
| Antibody | (Mouse monoclonal) Phospho-IκBα | Cell Signaling | Cat# 9246 | WB (1:1000) |
| Antibody | (Rabbit monoclonal) IκBα | Cell Signaling | Cat# 4812 | WB (1:1000) |
| Antibody | (Rabbit monoclonal) anti-phospho p44/42 (ERK1/2) | Cell Signaling | Cat# 4370 | WB (1:2000) |
| Antibody | (Rabbit monoclonal) anti-p44/42 (ERK1/2) | Cell Signaling | Cat# 4695 | WB (1:2000) |

*Appendix 1 Continued on next page*

*Appendix 1 Continued*

| Reagent type (species) or resource | Designation | Source or reference | Identifiers | Additional information |
|---|---|---|---|---|
| Antibody | (Rabbit monoclonal) anti-phospho SAPK/JNK | Cell Signaling | Cat# 4668 | WB (1:1000) |
| Antibody | (Rabbit monoclonal) anti-phospho AKT | Cell Signaling | Cat# 4060 | WB (1:1000) |
| Antibody | (Rabbit monoclonal) anti-phospho STAT3 | Cell Signaling | Cat# 9145 | WB (1:1000) |
| Antibody | (Mouse monoclonal) anti-SARS-CoV-2 S | GeneTex | Cat# GTX632604 | WB (1:1000) FACS (1 µl/ $1 \times 10^6$ cells) |
| Antibody | (Monoclonal) anti-CD16/CD32 | eBioscience | Cat# 14-0161-82 | FACS (1 µg/$1 \times 10^6$ cells) |
| Antibody | (Mouse monoclonal) anti-β-actin | Sigma-Aldrich | Cat# A2228 | WB (1:10,000) |
| Recombinant DNA reagent | GFP-Flag (plasmid) | VectorBuilder | Cat# VB200507-2985cmv | |
| Recombinant DNA reagent | SARS-CoV-2 S-Flag | VectorBuilder | Cat# VB200507-2984jyv | |
| Recombinant DNA reagent | TLR1 CRISPR/Cas9 | Santa Cruz | Cat# sc-423418 | |
| Recombinant DNA reagent | TLR2 CRISPR/Cas9 | Santa Cruz | Cat# sc-423981 | |
| Recombinant DNA reagent | TLR6 CRISPR/Cas9 | Santa Cruz | Cat# sc-423420 | |
| Sequence-based reagent | m_Il1b_F | This paper | PCR primers | GCCTCGTG CTGTCGG ACCCATA |
| Sequence-based reagent | m_Il1b_R | This paper | PCR primers | TGCAGGGT GGGTGTG CCGTCTT |
| Sequence-based reagent | m_Il6_F | This paper | PCR primers | CAA GAA AGA CAA AGC CAG AGT C |
| Sequence-based reagent | m_Il6_R | This paper | PCR primers | GAA ATT GGG GTA GGA AGG AC |
| Sequence-based reagent | m_Tnfa_F | This paper | PCR primers | TCCCAGGTTC TCTTCAAGGGA |
| Sequence-based reagent | m_Tnfa_R | This paper | PCR primers | GGTGAGGAG CACGTAGTCGG |
| Sequence-based reagent | m_Ifng_F | This paper | PCR primers | GAAAGACAA TCAGGCCATCA |
| Sequence-based reagent | m_Ifng_R | This paper | PCR primers | TTGCTGTTGC TGAAGAAGGT |
| Sequence-based reagent | m_Ifnb_F | This paper | PCR primers | GCCTGGATG GTGGTC CGAGCA |
| Sequence-based reagent | m_Ifnb_R | This paper | PCR primers | TACCAGTCC CAGAGTCC GCCTCT |
| Sequence-based reagent | m_Ifna_F | This paper | PCR primers | TCTGATGCA GCAGGTGGG |
| Sequence-based reagent | m_Ifna_R | This paper | PCR primers | AGGGCTCT CCAGACTTC TGCTCTG |
| Sequence-based reagent | m_Cxcl1_F | This paper | PCR primers | TGAGCTGCG CTGTCA GTGCCT |

*Appendix 1 Continued on next page*

*Appendix 1 Continued*

| Reagent type (species) or resource | Designation | Source or reference | Identifiers | Additional information |
|---|---|---|---|---|
| Sequence-based reagent | m_*Cxcl1*_R | This paper | PCR primers | AGAAGCCA GCGTTCA CCAGA |
| Sequence-based reagent | m_*Cxcl2*_F | This paper | PCR primers | CAA GAA CAT CCA GAG CTT GAG TGT |
| Sequence-based reagent | m_*Cxcl2*_R | This paper | PCR primers | GCC CTT GAG AGT GGC TAT GAC TT |
| Sequence-based reagent | h_*IL1B*_F | This paper | PCR primers | AAATACCTG TGGCCTTGGGC |
| Sequence-based reagent | h_*IL1B*_F | This paper | PCR primers | TTTGGGATC TACACTC TCCAGCT |
| Sequence-based reagent | h_*IL6*_F | This paper | PCR primers | GTAGCCGC CCCACACAGA |
| Sequence-based reagent | h_*IL6*_R | This paper | PCR primers | CATGTCTCCT TTCTCAG GGCTG |
| Sequence-based reagent | h_*TNFA*_F | This paper | PCR primers | CCCAGGGA CCTCTCT CTAATCA |
| Sequence-based reagent | h_*TNFA*_R | This paper | PCR primers | GCTTGAGGG TTTGCTA CAACATG |
| Sequence-based reagent | h_*IFNG*_F | This paper | PCR primers | CCAACGCAAA GCAATACATGA |
| Sequence-based reagent | h_*IFNG*_R | This paper | PCR primers | CCTTTTTCG CTTCCCT GTTTTA |
| Sequence-based reagent | h_*IFNB*_F | This paper | PCR primers | ATTGCCTCAA GGACAGGATG |
| Sequence-based reagent | h_*IFNB*_R | This paper | PCR primers | GGCCTTCA GGTAA TGCAGAA |
| Sequence-based reagent | h_*IFNA*_F | This paper | PCR primers | GTGAGGAAAT ACTTCCAAA GAATCAC |
| Sequence-based reagent | h_*IFNA*_R | This paper | PCR primers | TCTCATGAT TTCTGCTCT GACAA |
| Sequence-based reagent | h_*CXCL1*_F | This paper | PCR primers | AACCGAAGT CATAGCCACAC |
| Sequence-based reagent | h_*CXCL1*_R | This paper | PCR primers | CCTCCCTTC TGGTCAGTT |
| Sequence-based reagent | h_*CXCL2*_F | This paper | PCR primers | CGCCCAAAC CGAAGTCAT |
| Sequence-based reagent | h_*CXCL2*_R | This paper | PCR primers | GATTTGCCATT TTTCAG CATCTTT |
| Sequence-based reagent | h_*CCL2*_F | This paper | PCR primers | AGGTGACTGG GGCATTGAT |
| Sequence-based reagent | h_*CCL2*_R | This paper | PCR primers | GCCTCCAGCA TGAAAGTCTC |
| Peptide, recombinant protein | SARS-CoV-2 S1 | RayBiotech | Cat# 230-30161 | |
| Peptide, recombinant protein | SARS-CoV-2 S1 | R&D | Cat# 10569-CV-100 | |

*Appendix 1 Continued on next page*

*Appendix 1 Continued*

| Reagent type (species) or resource | Designation | Source or reference | Identifiers | Additional information |
|---|---|---|---|---|
| Peptide, recombinant protein | SARS-CoV-2 S2 | RayBiotech | Cat# 230-30163 | |
| Peptide, recombinant protein | SARS-CoV-2 S2 | R&D | Cat# 10594-CV-100 | |
| Peptide, recombinant protein | SARS-CoV-2 S-trimer | R&D | Cat# 10549-CV-100 | |
| Peptide, recombinant protein | SARS-CoV-2 N | RayBiotech | Cat# 230-30164 | |
| Peptide, recombinant protein | SARS-CoV-2 M | MyBioSource | Cat# MBS8574735 | |
| Peptide, recombinant protein | SARS-CoV-2 E | MyBioSource | Cat# MBS9141944 | |
| Commercial assay or kit | Pierce BCA Protein Assay Kit | Thermo Fisher Scientific | Cat# 23227 | |
| Commercial assay or kit | Mouse IL-6 ELISA Kit | R&D Systems | Cat# DY406-05 | |
| Commercial assay or kit | Mouse IL-1β ELISA Kit | R&D Systems | Cat# DY401-05 | |
| Commercial assay or kit | Mouse TNF-α ELISA Kit | R&D Systems | Cat# DY410-05 | |
| Commercial assay or kit | Human IL-6 ELISA Kit | R&D Systems | Cat# DY206-05 | |
| commercial assay or kit | Human IL-1β ELISA Kit | R&D Systems | Cat# DY201-05 | |
| Commercial assay or kit | Human TNF-α ELISA Kit | R&D Systems | Cat# DY210-05 | |
| Commercial assay or kit | SARS-CoV-2 S ELISA Kit | RayBiotech | Cat# ELV-COVID19S2 | |
| Chemical compound, drug | Phorbol-12-myristate 13-acetate (PMA) | InvivoGen | Cat# tlrl | 100 ng/ ml |
| Chemical compound, drug | TLR2-IN-C29 | Selleckchem | S6597 | 150 µM/ ml |
| Chemical compound, drug | Pam3CSK4 | InvivoGen | tlrl-pms | |
| Chemical compound, drug | FSL-1 | InvivoGen | tlrl-fsl | |
| Chemical compound, drug | ACE2 Inhibitor, MLN-4760 | Sigma-Aldrich | 5306160001 | 10 µM/ ml |
| Chemical compound, drug | Lipofectamine 3000 | Thermo Fisher Scientific | Cat# L3000015 | |
| Chemical compound, drug | Ultrapure *Escherichia coli*-derived LPS | InvivoGen | Cat# tlrl-smlps | |
| Software, algorithm | Flowjo v10 | Treestar, Inc | RRID:SCR_008520 | |
| Software, algorithm | CytoFLEX- | Beckman Coulter | | |
| Software, algorithm | GraphPad Prism | Graphpad.com | RRID:SCR_00279 | |

