## [Editor Report]

Your paper provides an important advance regarding the role of SARS-CoV-2 spike protein as an inducer of the innate inflammatory cascade in both epithelial cells and macrophages. This is an important early event in development of the cytokine storm associated with COVID-19 and may be of therapeutic relevance.

---

## [Decision Letter]

**Decision letter after peer review:**

Thank you for submitting your article "SARS-CoV-2 spike protein induces inflammation via TLR2-dependent activation of the NF-κB pathway" for consideration by *eLife*. Your article has been reviewed by 3 peer reviewers, and the evaluation has been overseen by a Reviewing Editor and Tadatsugu Taniguchi as the Senior Editor. The following individual involved in review of your submission has agreed to reveal their identity: Coy Allen (Reviewer #2).

Essential revisions:

1) The authors should discuss why their finding regarding TLR4 signaling appear to differ from those of other groups.

2) Repeat experiments with concurrent exposure to S1, S2, M, N, and E proteins would provide a more comprehensive picture of viral infection, and a nice comparison to S1 and S2 alone.

3) Experiments should be repeated in Calu-3 cells which are permissive to SARS-CoV-2.

4) The authors should probe the mechanisms of cytokine production in macrophage via TLR2.

*Reviewer #2 (Recommendations for the authors):*

As mentioned above, in general the studies are well done and the data supports the overall conclusions of the manuscript. However, the following comments should be considered in an effort to strengthen the work presented:

1) TLR2 signals as a heterodimer. Thus, it is not clear if the mechanism proposed is associated with the TLR1/2 or the TLR2/6 heterodimer. This is critical information necessary to confidently define the mechanism proposed.

2) A549 cells are refractory to SARS-CoV-2 infection (multiple sources, including The Lancet, 2020). Thus, their use here lacks significant physiological relevance. The epithelial cell studies should be conducted or at least confirmed in a more relevant cells line, such as the Calu-3 cells.

3) The direct connection between the TLR2/MyD88 mechanism in the mouse studies and the human cell line studies is lacking. The team states in the methods that they utilize C29 to attenuate TLR2 function in THP1s. However, the data provided is minimal. These studies should be repeated in all cell lines and include full panels of controls (such as polyI:C and PAM3) and evaluated across a time frame. A stronger study would be adding knockdown data in their cell lines to confirm the mouse studies.

*Reviewer #3 (Recommendations for the authors):*

The findings of this study are of potential importance. However, they are preliminary in nature and the biologic relevance of these findings remains unclear. A number of potential approaches could be taken to show biological relevance. Do macrophages infected with the SARS-CoV-2 virus induce cytokine production in a TLR2 dependent manner? Are human TLR2 polymorphisms associated with disease severity or response to vaccination? Is TLR2 required for effective vaccine responses for vaccines based on the spike protein?

---

## [Author Response]

Essential revisions:1) The authors should discuss why their finding regarding TLR4 signaling appear to differ from those of other groups.

Thank you for the suggestion. Two recent publications reported that spike protein is sensed by TLR4 (Shirato K, Heliyon, 2021; Zhao Y, Cell Res, 2021), while another paper showed that SARS-CoV-2 triggers inflammation via TLR2 (Zheng M, Nat Immunol, 2021). We believe that the discrepant results of those studies are possibly due to endotoxin contamination of the recombinant proteins used in those studies as they were generated in *E. coli*. To avoid this problem, we used recombinant proteins generated in mammalian HEK293T cells. We have now included a discussion on this issue in page 16.

2) Repeat experiments with concurrent exposure to S1, S2, M, N, and E proteins would provide a more comprehensive picture of viral infection, and a nice comparison to S1 and S2 alone.

As per recommendation, we have performed additional experiments in which we measured the effect of concurrent exposure of all structural proteins. The new data are now presented as Figure 1—figure supplement 1B. Our data suggest that other structural proteins do not have any positive or negative influence on S-induced inflammatory responses.

3) Experiments should be repeated in Calu-3 cells which are permissive to SARS-CoV-2.

We appreciate the reviewers’ and editor’s suggestion of repeating crucial experiments in Calu3 cells. We have now included a series of studies using Calu3 cells. The new studies are now presented as Figures 2B, 2D, 3C, 3F, 3H, Figure 3—figure supplement 1C and Figure 5—figure supplement 2.

4) The authors should probe the mechanisms of cytokine production in macrophage via TLR2.

We have demonstrated that S protein triggers inflammation via TLR2. Since TLR2 heterodimerizes with TLR1 or TLR6, it is a valid concern whether these coreceptors are involved in S-induced cytokine production. To address the reviewer’s concerns, we have used engineered HEK293T cells expressing TLR2, TLR2/TLR1or TLR2/TLR6 and showed that S protein can be sensed by both TLR2/1 and TLR2/6 heterodimers. Further, we knocked down TLR1, TLR2, TLR6 or TLR1 and TLR6 together in RAW264.7 macrophages. Following stimulation with S protein, we observed that cytokine production was abrogated when TLR2 and TLR1/6 were knocked out. These data suggest that upon ligation with S protein TLR2 in association with either TLR1 or TLR6 activates the downstream NF-κB pathway. Our new findings are shown in Figure 6, Figure 6—figure supplement 1 and Figure 6—figure supplement 2.

Reviewer #2 (Recommendations for the authors):As mentioned above, in general the studies are well done and the data supports the overall conclusions of the manuscript. However, the following comments should be considered in an effort to strengthen the work presented:

We greatly appreciate the reviewer’s insightful comments on this manuscript. We found that the reviewer’s comments and concerns were very constructive and helped us improve the quality of our studies. We have performed a series of additional studies to thoroughly address all of the concerns.

1) TLR2 signals as a heterodimer. Thus, it is not clear if the mechanism proposed is associated with the TLR1/2 or the TLR2/6 heterodimer. This is critical information necessary to confidently define the mechanism proposed.

The reviewer has pointed out an important concern. To understand whether heterodimerization with TLR1 and TLR6 is required for TLR2 in sensing S protein, we used HEK293T reporter cells overexpressing TLR2, TLR2/TLR1 or TLR2/TLR6. Interestingly, all three cells were responsive to S1, S2, and S-tri (Figure 6A-D). The inflammatory response of these reporter cells was inhibited when TLR2 was blocked by C29 (Figure 6E and F), suggesting that TLR2 is essential for sensing S protein. Given that HEK293T cells express low level of TLR1 and TLR6, but not TLR2, our data suggest that TLR2 heterodimerizes with either TLR1 or TLR6 in sensing S protein. To further verify this interpretation, we knocked out TLR2, TLR1, TLR6 or both TLR1 and TLR6 in the macrophage cell line. Following stimulation with S protein, we observed that cytokine production was abrogated in TLR2-KO and TLR1/6-double KO cells (Figure 6G, Figure 6—figure supplement 2). These data suggest that upon ligation with S protein TLR2 in association with either TLR1 or TLR6 activates the downstream NF-κB pathway.

2) A549 cells are refractory to SARS-CoV-2 infection (multiple sources, including The Lancet, 2020). Thus, their use here lacks significant physiological relevance. The epithelial cell studies should be conducted or at least confirmed in a more relevant cells line, such as the Calu-3 cells.

We appreciate the reviewer’s valuable suggestion. We have now included a series of studies using Calu3 cells. The new studies are now presented as Figures 2B, 2C, 3F, 3H, and Figure 3—figure supplement 1A and 1C, and Figure 5—figure supplement 2. These experiments demonstrated that, like A549 cells, Calu3 cells are stimulated by S proteins to produce cytokines and chemokines.

3) The direct connection between the TLR2/MyD88 mechanism in the mouse studies and the human cell line studies is lacking. The team states in the methods that they utilize C29 to attenuate TLR2 function in THP1s. However, the data provided is minimal. These studies should be repeated in all cell lines and include full panels of controls (such as polyI:C and PAM3) and evaluated across a time frame. A stronger study would be adding knockdown data in their cell lines to confirm the mouse studies.

We agree with the reviewers that additional controls should be included in studies where TLR2 was blocked by C29. We repeated the study including controls such as LPS and Pam3CSK4. Our data shown in Figure 5G confirms that C29 only inhibits the TLR2 pathway.

As per reviewer’s recommendation, we also evaluated the effect of C29 in Calu3 cells. Our data showed that C29 inhibits S-mediated cytokine production in Calu3 cells (Figure 5—figure supplement 2).

Finally, we show that RAW264.7 cells do not respond to S protein when TLR2 was knocked out with CRISPR/Cas9 (Figure 6G). All these data consistently and rigorously support the conclusion that TLR2 is the innate immune receptor for S protein and is involved in inflammatory response.

Reviewer #3 (Recommendations for the authors):The findings of this study are of potential importance. However, they are preliminary in nature and the biologic relevance of these findings remains unclear. A number of potential approaches could be taken to show biological relevance. Do macrophages infected with the SARS-CoV-2 virus induce cytokine production in a TLR2 dependent manner? Are human TLR2 polymorphisms associated with disease severity or response to vaccination? Is TLR2 required for effective vaccine responses for vaccines based on the spike protein?

We greatly appreciate the reviewers’ insightful comments and finding our study potentially important. We completely agree with the reviewer that showing inflammatory response of macrophage to live SARS-CoV2 is TLR2 dependent would make our finding more compelling. Due to the limitation of available resources for working with live SARS-CoV2, we were not able to perform such an experiment. However, a recent study has already addressed this concern showing that macrophages from *Tlr2^-/-^* mice are defective in cytokine production when infected with SARS-CoV-2 (Zheng M, Nat Immunol, 2021).

The reviewer’s comments on the role of TLR2 in vaccine response and an association of TLR2 polymorphism in severity of COVID19 are insightful and important. However, addressing these concerns is out of scope of this study, but we are working on these topics which will be reported in our future studies. We briefly discussed this issue in page 18.